# ATTRIBUTE-BASED INTERPRETABLE EVALUATION METRICS FOR GENERATIVE MODELS

## ABSTRACT

When the training dataset comprises a 1:1 proportion of dogs to cats, a generative model that produces 1:1 dogs and cats better resembles the training species distribution than another model with 3:1 dogs and cats. Can we capture this phenomenon using existing metrics? Unfortunately, we cannot, because these metrics do not provide any interpretability beyond "diversity". In this context, we propose a new evaluation protocol that measures the divergence of a set of generated images from the training set regarding the *distribution of attribute strengths* as follows. Single-attribute Divergence (SaD) reveals the attributes that are generated excessively or insufficiently by measuring the divergence of PDFs of individual attributes. Paired-attribute Divergence (PaD) reveals such pairs of attributes by measuring the divergence of *joint* PDFs of pairs of attributes. For measuring the attribute strengths of an image, we propose Heterogeneous CLIPScore (HCS) which measures the cosine similarity between image and text vectors with *heterogeneous initial points*. With SaD and PaD, we reveal the following about existing generative models. ProjectedGAN generates implausible attribute relationships such as `baby` with `beard` even though it has competitive scores of existing metrics. Diffusion models struggle to capture diverse colors in the datasets. The larger sampling timesteps of the latent diffusion model generate the more minor objects including `earrings` and `necklace`. Stable Diffusion v1.5 better captures the attributes than v2.1. Our metrics lay a foundation for explainable evaluations of generative models.

## 1 INTRODUCTION

The advancement of deep generative models, including VAEs (Kingma and Welling, 2013), GANs((Karras et al., 2019; 2020b; 2021; Sauer et al., 2021), and Diffusion Models (DMs) (Song et al., 2020; Nichol and Dhariwal, 2021; Rombach et al., 2022), has led to generated images that are nearly indistinguishable from real ones. Evaluation metrics, especially those assessing fidelity and diversity, play a pivotal role in this progress. One standout metric is Fréchet Inception Distance (FID) (Heusel et al., 2017), measuring the disparity between training and generated image distributions in embedding space. Coupled with other metrics like precision, recall, density, and coverage, the difference between generated and real image distributions is effectively gauged.

Figure 1 illustrates the evaluation metrics for two models with distinct properties. While Model 1's generated images align closely with the training dataset, Model 2 exhibits a lack of diversity. Notably, in Figure 1a gray box, Model 1 consistently outperforms Model 2 across all metrics. Yet, these metrics fall short in explicability; for example, they don't highlight the overrepresentation of `long hair` and `makeup` in Model 2.

Addressing this gap, our paper proposes a methodology to quantify discrepancies between generated and training images, focusing on specific attributes. Figure 1b shows the concept of our alternative approach that measures the distribution of attribute strengths compared to the training set: while Model 1 offers a balanced attribute distribution akin to the training dataset, Model 2 overemphasizes `long hair` and underrepresents `beard`.

To build metrics that quantify the difference between two image sets in an interpretable manner, we introduce Heterogeneous CLIPScore (HCS), an enhanced variant of CLIPScore (Radford et al., 2021). Compared to CLIPScore, Heterogeneous CLIPScore captures the similarity between modalities—image and text—by establishing distinct origins for text and image vectors.

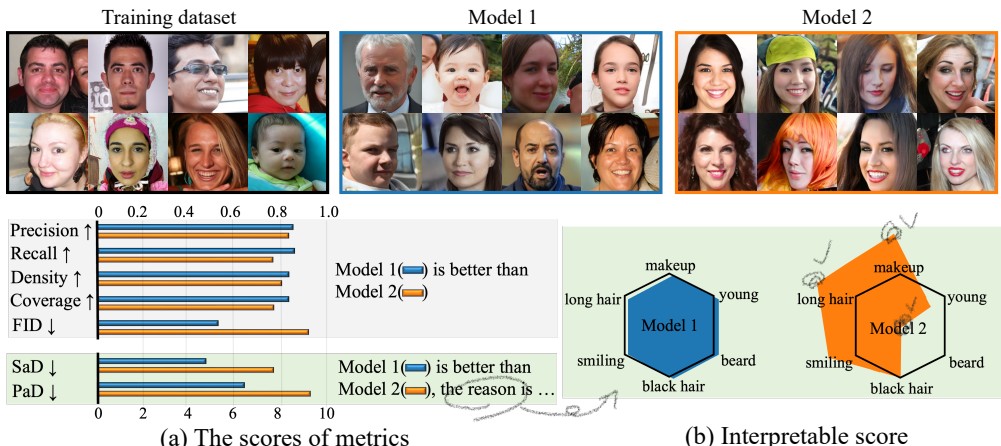

Figure 1: **Conceptual illustration of our metric.** We design the scenario, Model 2 lacks diversity. (a) Although existing metrics (gray box) capture the inferiority of Model 2, they do not provide an explanation for the judgments. (b) Our attribute-based proposed metric (green box) has an interpretation: Model 2 is biased regarding `long hair`, `makeup`, `smiling`, and `beard`.

Utilizing HCS, we introduce new evaluation protocols to assess the attribute distribution alignment between generated images and training data as follows. 1) Single-attribute Divergence (SaD) measures how much a generative model deviates from the distribution of each attribute in the training data. 2) Paired-attribute Divergence (PaD) measures how much a generative model breaks the relationship between attributes in the training data, such as "babies do not have beards." With the proposed metrics, users can now realize which specific attributes (or pairs of attributes) in generated images differ from those in training images.

Our protocols also enable flexible user-defined evaluation. Given our capability to assign each attribute, users can emphasize certain features without considering certain features, such as hair attributes (`long hair`, `black hair`, `blonde hair`), while excluding the apparent age like `baby` or `elderly`. Figure 1b shows SaD result with user-defined 6 attributes, where `long hair`, `makeup`, `beard` are the most influential attributes to SaD. We note elaborate quantification of attribute preservation could be one of the meaningful tasks since the generative model can be utilized for diverse purposes such as text-to-image generation not only for generating a plausible image.

We conduct a series of carefully controlled experiments with varying configurations of attributes to validate our metrics in Section 5.1 and 5.2. Then we provide different characteristics of state-of-the-art generative models (Karras et al., 2019; 2020b; 2021; Sauer et al., 2021; Nichol and Dhariwal, 2021; Rombach et al., 2022; Yang et al., 2023) which could not be seen in the existing metrics. For instance, GANs better synthesize color-/texture-related attributes such as `striped fur` which DMs hardly preserve in LSUN-Cat (Section 5.3). When we increase the sampling steps of DMs, tiny objects such as `necklaces` and `earrings` tend to appear more frequently. Even though Stable diffusion v2.1 is reported that have a better FID score than Stable diffusion v1.5, the attribute-aspect score is worse than v1.5 (Section 5.4). Our approach is versatile, and applicable wherever image comparisons are needed. The code will be publicly available.

## 2 RELATED WORK

**Fréchet Inception Distance** Fréchet Inception Distance (FID) (Heusel et al., 2017) calculates the distance between the estimated Gaussian distributions of two datasets using a pre-trained Inception-v3 (Szegedy et al., 2016). However, Kynkäänniemi et al. (2022) noted issues with embeddings when generated images deviate significantly from training data. This led to proposals of using CLIP (Radford et al., 2021) encoder, which aligns text and images in a shared space, instead of Inception-v3. However, they directly use the raw embedding of CLIP encoder while we design a new representation.

**Fidelity and diversity** Sajjadi et al. (2018) devised precision and recall for generative model evaluation. Further refinements were provided by Kynkäänniemi et al. (2019) and Naeem et al. (2020).

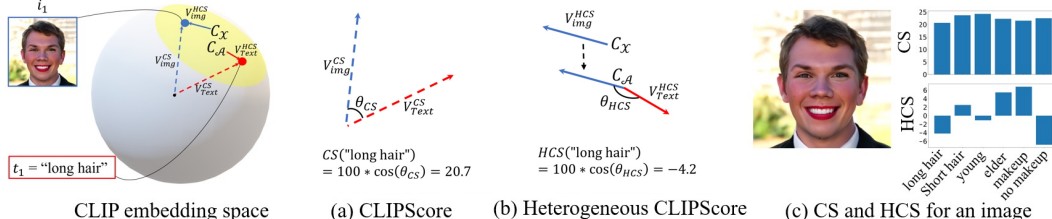

Figure 2: **Illustration of CLIPScore and Heterogeneous CLIPScore.** (a) CLIPScore (CS) evaluates the similarity between $V_{img}^{CS}$ and $V_{Text}^{CS}$ from the coordinate origin, where the angle between the two vectors is bounded, resulting in a limited similarity value. (b) Heterogeneous CLIPScore (HCS) gauges the similarity between $V_{img}^{HCS}$ and $V_{Text}^{HCS}$ using the defined means of images $C_{\mathcal{X}}$ and texts $C_{\mathcal{A}}$ as the origin, the range of similarity is unrestricted. (c) shows flexible values of HCS compared to CS.

Generally, these metrics use a pre-trained network to evaluate how embeddings of generated images match with those of real images and vice-versa.

**Other metrics** Beyond these, metrics such as Perceptual path length (Karras et al., 2019), Fréchet segmentation distance (Bau et al., 2019), and Rarity score (Han et al., 2022) have been introduced. The first indicates latent space smoothness, the second measures pixel segmentation differences, and the latter assesses the rarity of generated images. However, these metrics predominantly rely on raw embeddings from pretrained classifiers, yielding scores with limited interpretability. As Figure 1a indicates, while some metrics highlight poor image generation performance, they lack in-depth explanatory insights. We aim to fill this gap with our novel, detailed, and insightful evaluation metrics.

TIFA (Hu et al., 2023) uses visual question answering to validate if text-to-image results correspond to the input texts. On the other hand, our metrics evaluate the distribution of attribute strengths in a set of images.

## 3 TOWARD EXPLAINABLE METRICS

Existing metrics for evaluating generated images often use embeddings from Inception-V3 (Szegedy et al., 2016) or CLIP image encoder (Dosovitskiy et al., 2020). Yet, these embeddings lack clarity in interpreting each channel in the embedding. Instead, we opt to measure attribute strengths in images for a predefined set of attributes. We first explain CLIPScore as our starting point (Section 3.1), introduce Heterogeneous CLIPScore (Section 3.2), and describe ways of specifying the target attributes (Section 3.3.)

### 3.1 MEASURING ATTRIBUTE STRENGTHS WITH CLIP

For a set of attributes, we start by measuring the attribute strengths of images. The typical approach is computing CLIPScore:

$$\text{CLIPScore}(x, a) = 100 * \text{sim}(\mathbf{E_I}(x), \mathbf{E_T}(a)), \tag{1}$$

where $x$ is an image, $a$ is a given text of attribute, $\text{sim}(*, *)$ is cosine similarity, and $\mathbf{E_I}$ and $\mathbf{E_T}$ are CLIP image encoder and text encoder, respectively. Figure 2c shows an example CLIPScores of an image regarding a set of attributes. Yet, CLIPScores themselves do not provide a clear notion of attribute strengths as we observe ambiguous similarities between opposite attributes. The research community is already aware of such a problem. To overcome this, we introduce Heterogeneous CLIPScore in the subsequent subsections, showcased in Figure 2c, ensuring more accurate attribute strengths.

Table 1: **CLIPScore and Heterogeneous CLIPScore's accuracy on CelebA dataset.**

|  | accuracy | f1 score |
|---|---|---|
| Heterogeneous CLIPScore | **0.817** | **0.616** |
| CLIPScore | 0.798 | 0.575 |

## 3.2 HETEROGENEOUS CLIPSCORE

In the earlier section, we noted that CLIPScore tends to have a narrow value range, as visualized in Figure 2a. To remedy this, we introduce Heterogeneous CLIPScore (HCS). It uses heterogeneous initial points for image and text embedding vectors as follows.

Given training images denoted as $\{x_1, x_2, ..., x_{N_\mathcal{X}}\} \in \mathcal{X}$, and a set of attributes defined as $\{a_1, a_2, ..., a_{N_\mathcal{A}}\} \in \mathcal{A}$, we define $C_\mathcal{X}$ as the center of images and $C_\mathcal{A}$ as another center of text attributes on CLIP embedding, respectively as

$$C_\mathcal{X} = \frac{1}{N_\mathcal{X}} \sum_{i=1}^{N_\mathcal{X}} \mathbf{E_I}(x_i), \quad C_\mathcal{A} = \frac{1}{N_\mathcal{A}} \sum_{i=1}^{N_\mathcal{A}} \mathbf{E_T}(a_i). \tag{2}$$

These centers act as initial points of the embedding vectors. HCS is defined by the similarity between the two vectors, $V_x$ and $V_a$. The former connects the image center to a specific image, while the latter connects the attribute center to a particular attribute. Then we define

$$V_x = \mathbf{E_I}(x) - C_\mathcal{X}, \quad V_a = \mathbf{E_T}(a) - C_\mathcal{A}, \tag{3}$$

$$\text{HCS}(x, a) = 100 * \text{sim}(V_x, V_a), \tag{4}$$

where $\text{sim}(*, *)$ computes cosine similarity. For extending HCS from a single sample to all samples, we denote the probability density function (PDF) of $\text{HCS}(x_i, a_i)$ for all $x_i \in \mathcal{X}$ as $\text{HCS}_\mathcal{X}(a_i)$.

Figure 2 illustrates the difference between HCS (Heterogeneous CLIPScore) and CS (CLIPScore). HCS uses the respective centers as initial points, allowing for clearer determination of attribute magnitudes, whereas CS lacks this clarity.

HCS also outshines CS in classifying attributes as shown in Table 1. This table displays the accuracy of CS and HCS using ground truth attributes in CelebA (Liu et al., 2015). Accuracy is computed by performing binary classification on all CelebA attributes using CS and HCS and comparing them to ground truth labels. HCS consistently surpasses CS. This accuracy trend persists even for refined attributes, excluding subjective ones such as `Attractive` or `Blurry`. The full accuracy, including `Attractive` and `Blurry`, is in Table S8. More details are available in the Appendix A.2.

## 3.3 ATTRIBUTE SELECTION

The effectiveness of our evaluation metric is contingent upon the target attributes we opt to measure. To determine the best attributes that truly capture generator performance, we put forth two methods for attribute selection.

**Caption-extracted attributes**  Our goal is to pinpoint and assess the attributes evident in the training data via image descriptions. By analyzing the frequency of these attributes in image captions, we can identify which ones are most prevalent. To achieve this for captionless datasets, we employ the image captioning model, BLIP (Li et al., 2022), to extract words related to attributes from the training data. We then adopt $N$ frequently mentioned ones as our target attributes, denoted as $\mathcal{A}$, for the metric. Given that these attributes are derived automatically, utilizing BLIP for this extraction could serve as a foundational method. Nevertheless, our approach retains flexibility for user-defined inputs as follows.

**User annotation**  Another method for attribute selection involves utilizing human-annotated attributes. By directly choosing attributes for evaluating generative models, users can compare the influence of each attribute score or select specific attributes for a particular purpose. Notably, CelebA

offers annotated attributes, serving as a good example of this approach. While external models such as GPT-3 (Brown et al., 2020) can aid in selecting a large number of attributes, it is important to use external models judiciously, given the potential for bias in the attributes it extracts. For an example of using GPT-3, see Appendix A.1.

# 4 EVALUATION METRICS WITH ATTRIBUTE STRENGTHS

In this section, we harness the understanding of attribute strengths to devise two comprehensible metrics. Section 4.1 introduces Single-attribute Divergence (SaD), quantifying the discrepancy in attribute distributions between training data and generated images. Section 4.2 brings forth Paired-attribute Divergence (PaD), evaluating the relationship between attribute strengths.

## 4.1 SINGLE-ATTRIBUTE DIVERGENCE

If we have a dataset with dogs and cats, and a generative model only makes dog images, it is not an ideal model because it does not produce cats at all (Goodfellow et al., 2016). With this idea, we say one generative model is better than another if it makes a balanced number of images for each attribute similar to the training dataset. Since we do not know the true distribution of real and fake images, we came up with a new metric, Single-attribute Divergence (SaD). This metric checks how much of each attribute is in the dataset by utilizing interpretable representation. Our metric, SaD, quantifies the difference in density for each attribute between the training dataset ($\mathcal{X}$) and the set of generated images ($\mathcal{Y}$). We define SaD as

$$\text{SaD}(\mathcal{X}, \mathcal{Y}) = \frac{1}{M} \sum_i^M \text{KL}(\text{HCS}_{\mathcal{X}}(a_i), \text{HCS}_{\mathcal{Y}}(a_i)), \tag{5}$$

where $i$ denotes an index for each attribute, $M$ is the number of attributes, KL(*) is Kullback-Leibler divergence, and $\text{HCS}_{\mathcal{X}}(a_i)$ denotes PDF of $\text{HCS}(x_i, a_i)$ for all $x_i \in \mathcal{X}$.

We analyze PDFs of Heterogeneous CLIPScore for each attribute present in $\mathcal{X}$ and $\mathcal{Y}$. These HCS PDFs reflect the distribution of attribute strengths within datasets. If an attribute's distribution in $\mathcal{X}$ closely mirrors that in $\mathcal{Y}$, their respective HCS distributions will align, leading to similar PDFs. To measure discrepancies between these distributions, we employ Kullback-Leibler Divergence (KLD). This quantifies how much the generated images either over-represent or under-represent specific attributes compared to the original data. Subsequently, we determine the average divergence across all attributes between $\mathcal{X}$ and $\mathcal{Y}$ to derive the aggregated metric for SaD.

In addition, we define the mean difference of attribute strength to further examine whether poor SaD comes from excessive or insufficient strength of an attribute $a$:

$$\text{mean difference} = \frac{1}{N_x} \sum_i^{N_x} \text{HCS}(x_i, a) - \frac{1}{N_y} \sum_i^{N_y} \text{HCS}(y_i, a). \tag{6}$$

where $N_x$ and $N_y$ are the number of training images and generated images, respectively. Intuitively, a high magnitude of mean difference indicates the mean strength of $\mathcal{Y}$ differs significantly from $\mathcal{X}$ for attribute $a$. A positive value indicates $\mathcal{Y}$ has images with stronger $a$ than $\mathcal{X}$, and vice versa for a negative value. While this does not conclusively reveal the exact trend due to $a$'s complex distribution, it provides an intuitive benchmark.

## 4.2 PAIRED-ATTRIBUTE DIVERGENCE

We introduce another metric, Paired-attribute Divergence (PaD), aimed at evaluating whether generated images maintain the inter-attribute relationships observed in the training data. Essentially, if specific attribute combinations consistently appear in the training data, generated images should also reflect these combinations. To illustrate, if every male image in the training dataset is depicted wearing glasses, the generated images should similarly represent males with glasses. We assess this by examining the divergence in the joint probability density distribution of attribute pairs between the training data and generated images. This metric, termed Paired-attribute Divergence (PaD), leverages

joint probability density functions as detailed below:

$$\text{PaD}(\mathcal{X}, \mathcal{Y}) = \frac{1}{|P|} \sum_{(i,j)}^{P} \text{KL}(\text{HCS}_{\mathcal{X}}(a_{i,j}), \text{HCS}_{\mathcal{Y}}(a_{i,j})), \tag{7}$$

where $M$ is the number of attributes, $P = \binom{M}{2}$, $(i,j)$ denotes an index pair of attributes selected out of $M$, and the joint PDF of the pair of attributes is denoted as $\text{HCS}_{\mathcal{X}}(a_{i,j})$.

When utilized together with SaD, PaD will offer a comprehensive analysis of the model's performance. For instance, if the probability density function of the generator for the attribute pair (`baby`, `beard`) diverges notably from the training data's distribution while SaD for `baby` and `beard` are comparatively low, it suggests that the generator may not be effectively preserving the (`baby`, `beard`) relationship. Consequently, PaD enables us to quantify how well attribute relationships are maintained in generated data. Moreover, it facilitates the measurement of attribute interdependencies, an aspect not extensively addressed in prior studies.

## 5 EXPERIMENTS

**Experiment details**   For estimating the probability density function (PDF) of Heterogeneous CLIPScore (HCS) in both the training data and generated images, Gaussian kernel density estimation is employed. We extract 10,000 samples from generated and real images to obtain PDFs of attribute strengths, which are then used to compute SaD and PaD. In every experiment, we use a set of $N_{\mathcal{A}} = 20$ attributes. In the case of FFHQ, USER attributes from CelebA ground truth were used.

### 5.1 BIASED DATA INJECTION EXPERIMENT: THE EFFECTIVENESS OF OUR METRIC

In this subsection, we conduct a toy experiment to validate our metrics against existing methods. Initially, two non-overlapping subsets, each with 30K images from FFHQ, are defined as training data $\mathcal{X}$ and generated images $\mathcal{Y}$. Starting with these subsets that share a similar distribution, we gradually infuse biased data into $\mathcal{Y}$. The biased data is generated using DiffuseIT (Kwon and Ye, 2022). We translate samples from the training data, without overlap to the initial 60K images, into `makeup` (Figure 5.1a) and `bangs` (Figure 5.1b). We also provide controlled counterpart where injected samples are unbiased data translated into the `person` (Figure 5.1c), or injected samples remain untranslated (Figure 5.1d).

As depicted in Figure 5.1, our metrics display a consistent trend: SaD and PaD rise with the inclusion of more edited images in $\mathcal{Y}$, whereas other metrics are static. Thanks to the attribute-based design, our metric suggests that `makeup` or `bangs` is the dominant factor for SaD, and relationships that are rarely seen in training data such as (`man`, `makeup`) and (`man`, `bangs`) for PaD. The impact on SaD and PaD scales linearly with the number of images from different attribute distributions. For an expanded discussion and additional experiments, refer to Figure S7 and Appendix B.3. These results underscore that SaD adeptly discerns the attribute distribution variation, and PaD identifies the joint distribution shift between attribute pairs, outperforming other metrics.

### 5.2 DISCERNMENT OF PAD

In another toy experiment, we designed a scenario where SaD metric struggled to detect specific attribute relationships, while PaD metric successfully pinpointed them. We used curated CelebA subsets as training data $\mathcal{X}$ and generated images $\mathcal{Y}$, ensuring discrepancies in attribute relationships.

For $\mathcal{X}$, we gathered 20,000 `smiling men` and 20,000 `non-smiling women` using CelebA's ground truth labels. In contrast, $\mathcal{Y}$ comprised 20,000 `non-smiling men` and 20,000 `smiling women`. While we cannot get insights into attribute relationship errors through exploring SaD (Figure 5.2a), examining PaD (Figure 5.2b) provides us with valuable clues.

Figure 5.2b highlights divergence of (`woman`, `smiling`) and (`man`, `smiling`) notably influence PaD. These findings demonstrate the superior sensitivity and discernment of our proposed metrics, allowing for a more comprehensive evaluation of generative models. For example, PaD of Project-edGAN (Sauer et al., 2021) is higher than other state-of-the-art generative models as shown in Table 2

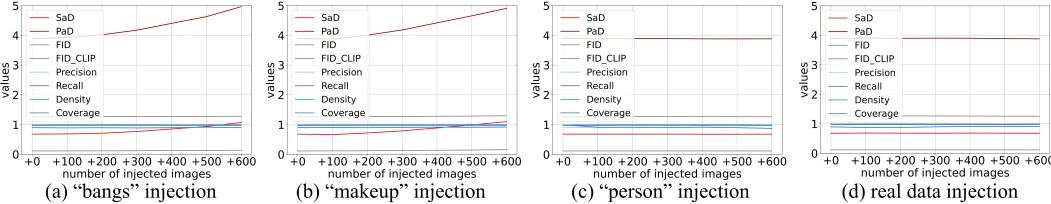

(a) "bangs" injection  (b) "makeup" injection  (c) "person" injection  (d) real data injection

Figure 3: **Validation of metrics through biased injection.** We design one set: typical 30k of FFHQ images, and another set: 30k FFHQ + injected images. Biased data injection, illustrated in (a) with `makeup` and (b) with `bangs` leads to an increase in both SaD and PaD rise. In contrast, unbiased data injection (c) `person` and (d) real data, injecting the same distribution as the training set results in no SaD and PaD rise. Our metrics effectively capture changes in attribute distribution, while existing metrics cannot.

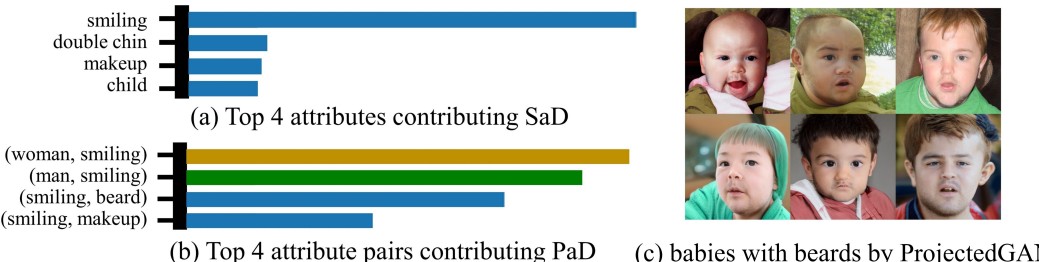

(a) Top 4 attributes contributing SaD

(b) Top 4 attribute pairs contributing PaD    (c) babies with beards by ProjectedGAN

Figure 4: **Necessity of PaD.** We define curated subsets of CelebA-HQ as training images, consisting of `smiling men` and `non-smiling women`, and generated images, consisting of `non-smiling men` and `smiling women`. (a) While SaD only specifies problematic attributes, (b) PaD identifies problematic attribute pairs such as (`woman`, `smiling`). (c) ProjectedGAN disregards attribute relationships, such as generating babies with beards.

Table 2: **Comparing the performance of generative models.** We computed each generative model's performance on our metric with their official pretrained checkpoints on FFHQ (Karras et al., 2019). We used 50,000 images for both GT and the generated set. We used USER attributes for this experiment.

| | StyleGAN1 | StyleGAN2 | StyleGAN3 | iDDPM | LDM (50) | LDM (200) | StyleSwin | ProjectedGAN |
|---|---|---|---|---|---|---|---|---|
| SaD $(10^{-7})\downarrow$ | 11.35 | **7.52** | 7.79 | 14.78 | 10.42 | 14.04 | 10.76 | 17.61 |
| PaD $(10^{-7})\downarrow$ | 27.25 | **19.22** | 19.73 | 34.04 | 25.36 | 30.71 | 26.56 | 41.53 |
| FID$\downarrow$ | 4.74 | **3.17** | 3.20 | 7.31 | 12.18 | 11.86 | 4.45 | 5.45 |
| FID$_{\text{CLIP}}\downarrow$ | 3.17 | **1.47** | 1.66 | 2.39 | 3.89 | 3.57 | 2.45 | 3.63 |
| Precision$\uparrow$ | 0.90 | 0.92 | 0.92 | 0.93 | **0.94** | 0.91 | 0.92 | 0.92 |
| Recall$\uparrow$ | 0.86 | 0.89 | 0.90 | 0.84 | 0.82 | 0.88 | 0.91 | **0.92** |
| Density$\uparrow$ | 1.05 | 1.03 | 1.03 | **1.09** | 1.09 | 1.07 | 1.01 | 1.05 |
| Coverage$\uparrow$ | 0.97 | 0.97 | 0.97 | 0.95 | 0.94 | 0.97 | 0.97 | 0.97 |

and we observe there are implausible attribute relationships such as (`baby`, `beard`) as shown in Figure 5.2b. We will discuss this in detail in the following Section 5.3.

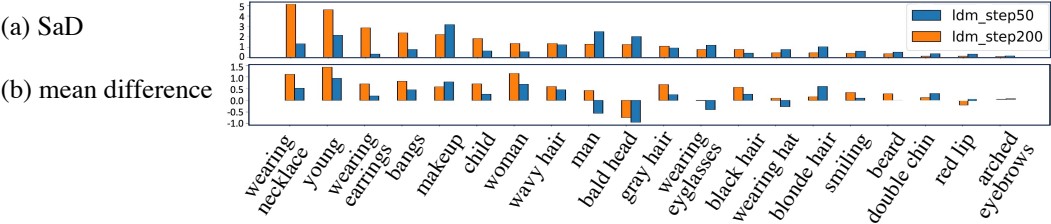

(a) SaD

(b) mean difference

Figure 5: **LDM with 50 steps v.s. LDM with 200 timesteps.** With increased sampling timesteps, (a) SaD of LDM gets worse, (b) since making too many fine objects such as `earrings` or `necklace`.

Table 3: **SaD and PaD of models with different attributes for LSUN Cat.** Analyzing the weakness of iDDPM for specific attribute types, such as color or shape. We used GPT-extracted attributes for this experiment.

| | color attributes | | shape attrbutes | |
|---|---|---|---|---|
| | SaD $(10^{-7})\downarrow$ | PaD $(10^{-7})\downarrow$ | SaD $(10^{-7})\downarrow$ | PaD $(10^{-7})\downarrow$ |
| StyleGAN1 (Karras et al., 2019) | **139.03** | **248.96** | 169.76 | 318.46 |
| StyleGAN2 (Karras et al., 2020b) | **112.06** | **195.75** | 132.41 | 246.44 |
| iDDPM (Nichol and Dhariwal, 2021) | 46.93 | 85.99 | **32.48** | **62.69** |

## 5.3 COMPARING GENERATIVE MODELS WITH OUR METRICS

Leveraging the superior sensitivity and discernment of our proposed metrics, we evaluate the performance of GANs and Diffusion Models (DMs) in Table 2. Generally, the tendency of SaD and PaD align with other existing metrics. However three notable points emerge; 1) ProjectedGAN (Sauer et al., 2021) lags in performance, 2) As sampling timesteps in DM increase, FIDs improve, while SaD and PaD decline. 3) GANs and Diffusion models vary in their strengths and weaknesses concerning specific attributes.

1) ProjectedGAN (Sauer et al., 2021) prioritizes matching the training set's embedding statistics for improving FID rather than improving actual fidelity (Kynkäänniemi et al., 2022). While it performs well in existing metrics, it notably underperforms in SaD and particularly in PaD. This implies that directly mimicking the training set's embedding stats does not necessarily imply correct attribute correlations. Figure 5.2b provides failure cases generated by ProjectedGAN.

2) Diffusion models typically yield better quality with higher number of sampling timesteps. Yet, SaD and PaD scores for LDM with 200 steps surpass those of LDM with 50 steps. As illustrated in Figure 5, higher sampling timesteps in the LDM model produce more high-frequency elements such as `necklaces` and `earrings`. This could explain the dominance of attributes such as `young`, `makeup`, `woman`, `wavy hair` naturally. We suppose that dense sampling trajectory generates more high-frequency objects. The scores and mean differences of each attribute are depicted in Figure 5a and Figure 5b respectively.

In addition, iDDPM shows notable scores, with the attribute `arched eyebrows` showing scores over two times higher than GANs in SaD, and attributes related to `makeup` consistently receive high scores across all StyleGAN 1, 2, and 3 models in PaD. Investigating how the generation process of GANs or DMs affects attributes such as attributes would be an intriguing avenue for future research. See Appendix C for details.

3) Diffusion models fall short on modeling color-related attributes than shape-related attributes. As our metrics provide flexible customization, we report SaD and PaD of color attributes (e.g., `yellow fur`, `black fur`) and shape attributes (e.g., `pointy ears`, `long tail`) within LSUN Cat dataset. Table 3 shows that iDDPM excels in matching shape attributes compared to color attributes. This aligns with the hypothesis by Khrulkov et al. (2022) suggesting that DMs learn the Monge optimal transport map, the shortest trajectory, from Gaussian noise distribution to image distribution regardless of training data. This implies that when the initial latent noise $x_T$ is determined, the image color is also roughly determined because the diffused trajectory tends to align with the optimal transport map.

## 5.4 EVALUATING TEXT-TO-IMAGE MODELS

Recently, there has been a huge evolution of text-to-image generative models (Nichol et al., 2021; Rombach et al., 2022; Saharia et al., 2022; Balaji et al., 2022). To evaluate text-to-image models, zero-shot FID score on COCO (Lin et al., 2014) is widely used including Stable Diffusion (SD). Instead, we use our metrics to examine text-to-image models regarding excessively or insufficiently generated attributes. We generate 30K images with captions from COCO using SDv1.5 and SDv2.1 to calculate SaD and PaD with attributes extracted from the captions. We use $N_A = 30$.

Table 4 shows SDv1.5 has twice better SaD and PaD than SDv2.1. Interestingly, the mean difference of attribute strengths is below zero. It implies that SDs tend to omit some concepts such as `group`[1]

---

[1]e.g., A group of people is standing around a large clock.

Table 4: **SaD and PaD of different versions of Stable Diffusion.** Stable Diffusion v1.5 is almost twice better than v2.1. We generate 30k images using the captions from COCO. We use $N_{\mathcal{A}} = 30$.

| $N_{\mathcal{A}} = 30$ | SaD $(10^{-7})\downarrow$ | PaD $(10^{-7})\downarrow$ | SaD worst-rank attr (mean difference) | | |
|---|---|---|---|---|---|
| | | | 1st | 2nd | 3rd |
| SDv1.5 | 24.37 | 60.71 | plate (-1.9) | group (-1.6) | building (-1.6) |
| SDv2.1 | 48.23 | 106.86 | group (-3.7) | plate (-2.5) | person (-2.7) |

(a) number of samples          (b) number of attributes

Figure 6: **SaD and PaD over a different number of samples and attributes.** (a) SaD and PaD are stable with more than 50,000 images. (b) The ranking of models mostly remains consistent regardless of the number of attributes.

or `plate`[2]. In particular, SDv2.1 struggles to generate scenes with multiple people. It aligns with common claims[3] about SDv2.1 even though it achieves low FID. We provide more details in Appendix B.4.

## 5.5 IMPACT OF SAMPLE SIZE AND ATTRIBUTE COUNT ON PROPOSED METRIC

In Figure 6, we conduct ablation experiments to study the impact of the number of samples and attributes. Using four random seeds, we generate images with StyleGAN3 from FFHQ. We posit that SaD and PaD begin to standardize with 30,000 images and become more stable with over 50,000 images. Figure 6b provides SaD and PaD of various models over different numbers of attributes where the attributes from BLIP are sorted by their number of occurrences in the dataset. The ranking of the models largely stays stable irrespective of the number of attributes. However, the rank of LDM rises as rarely occurring attributes are included, as depicted by the purple line in Figure 6b. The rare attributes are `scarf, flower, and child`. We suggest that 20 attributes are sufficient for typical evaluation, but leveraging a broader range offers richer insights.

## 6 CONCLUSION AND DISCUSSION

We have introduced novel metrics that evaluate the distribution of attribute strengths. Single-attribute Divergence reveals which attributes are correctly or incorrectly modeled. Paired-attribute Divergence considers the joint occurrence of attributes in individual images. The explicit interpretability of these metrics allows us to know which generative model suits the user's necessity. Furthermore, Heterogeneous CLIPScore more accurately captures the attribute strengths than CLIPScore.

Our metrics have the advantage of revealing the distribution of attributes from a *set* of generated images where human judgment faces difficulty in observing attributes in excessively many images. Furthermore, our research establishes a solid foundation for the development of explainable evaluation metrics for generative models and contributes to the advancement of the field.

**Discussion**   1) Estimating PDFs with KDE requires a sufficient (>50K) number of samples. 2) Our metrics can be influenced by quality of attribute detector. 3) While our metrics are highly customizable with different sets of attributes, the target attributes should be chosen to meet the users' expectations. I.e., a limited or biased set of attributes might mislead our metrics. 4) Exploring strengths of other aspects such as texture (Caron et al., 2021; Oquab et al., 2023; Kirillov et al., 2023) or other modalities (Girdhar et al., 2023) may provide valuable insights and enhance the robustness of our metrics.

---

[2]e.g., A table is set with two plates of food and a candle.
[3]https://www.assemblyai.com/blog/stable-diffusion-1-vs-2-what-you-need-to-know/

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

# 7 APPENDIX

# A IMPLEMENTATION DETAILS

## A.1 ADDITIONAL EXPERIMENT SETUP

**Details of generated images**   We generate samples using official checkpoints provided by Style-GANs (Karras et al., 2019; 2020b;a; 2021), ProjectedGAN (Sauer et al., 2021), Styleswin (Zhang et al., 2022), iDDPMs (Nichol and Dhariwal, 2021; Choi et al., 2022), and LDM (Rombach et al., 2022). We use 50k of training images and generated images for both FFHQ (Karras et al., 2019) and LSUN Cat (Yu et al., 2015) experiment.

**Details of GPT queries**   Table S5 provides the questions we used for preparing GPT attributes. We accumulated GPT attributes by iteratively asking GPT to answer 'Give me 50 words of useful, and specific adjective visual attributes for {*question*}'. Then, we selected the top N attributes based on their frequency of occurrence, ensuring that the most frequently mentioned attributes were prioritized. We suppose that the extracted attributes might be biased due to the inherent randomness in GPT's answering process. This potential problem is out of our scope. We anticipate future research will address it to extract attributes in a more fair and unbiased manner with large language models. For a smooth flow of contents, the table is placed at the end of this material.

Table S5: **Scripts used for extracting attributes from GPT.** We stack GPT attributes by iteratively asking GPT to answer 'Give me 50 words of useful, and specific adjective visual attributes for {*question*}'.

| Dataset | *question* |
|---------|-----------|
| FFHQ | 'distinguishing faces in a photo' |
| | 'distinguishing human faces in a photo' |
| | 'distinguishing different identities of people in photos of faces' |
| | 'differentiating between people's faces by their distinctive features' |
| | 'people to change there styles in hairs, accessories around their faces' |
| | 'recognizing changes in hair and accessory styles in photographs of people's faces' |
| | 'identifying distinct faces within an image |
| | 'recognizing facial characteristics to distinguish people in photos' |
| | 'discerning variations in facial features to identify people in images' |
| | 'spotting differences in facial appearance for identifying individuals' |
| LSUN Cat | ' recognizing individuals from facial features in photographs' |
| | 'identifying distinct faces within an image |
| | 'recognizing variations in feline appearance to identify individual cats' |
| | 'discerning differences in fur patterns and colors to distinguish cats in photos' |
| | 'detecting subtle facial expressions to distinguish emotions in cat photos' |
| | 'differentiating between cats based on body type and size in photos' |
| | 'identifying distinctive facial features to distinguish between cats in images' |
| | 'recognizing changes in coat texture and length in photos of cats' |
| | 'discerning variations in eye color and shape to identify individual cats in images' |
| | 'spotting unique markings to distinguish between cats in photos' |

**Details of extracted attribute**   Table S6 describes selected attributes by each extractor. We used "A photo of {attribute}" as prompt engineering for all attributes.

**Miscellaneous**   We use `scipy.stats.gaussian_kde`(dataset, '*scott*', None) to estimate the distribution of Heterogeneous CLIPScore for given attributes. We use `spacy.load("en_core_web_sm")` to extract attributes from BLI P(Li et al., 2022) captions. We resize all images to 224x224. We used `"ViT-B/32"` (Dosovitskiy et al., 2020) as a CLIP encoder. We used a single NVIDIA RTX 3090 GPU (24GB) for the experiments.

Table S6: **Examples of attributes for each attribute extractor.**

| Extractor | N | Attribute |
|---|---|---|
| BLIP | 20 | woman, man, person, glasses, suit, little girl, tie, picture, sunglasses, young boy, cell phone, microphone, necklace, hat, young girl, blonde hair, long hair, blue shirt, beard, white shirt |
| | 30 | woman, man, person, glasses, suit, little girl, tie, picture, sunglasses, young boy, cell phone, microphone, necklace, hat, young girl, blonde hair, long hair, blue shirt, beard, white shirt, her head, her face, couple, baby, her hair, scarf, black shirt, smile, young man, little boy, child |
| | 40 | woman, man, person, glasses, suit, little girl, tie, picture, sunglasses, young boy, cell phone, microphone, necklace, hat, young girl, blonde hair, long hair, blue shirt, beard, white shirt, her head, her face, couple, baby, her hair, scarf, black shirt, smile, young man, little boy, child, red hair, flower, her hand, his mouth, blue eyes, women |
| GPT | 20 | clean-shaven, beard, mustache, wide-eyed, thin lips, bald, glasses-wearing, freckled, almond-shaped eyes, scarred, wrinkled, soul patch, high forehead, hooded eyes, piercings, prominent cheekbones, full lips, braided, upturned-nosed, youthful |
| | 30 | clean-shaven, beard, mustache, wide-eyed, thin lips, bald, glasses-wearing, freckled, almond-shaped eyes, scarred, wrinkled, soul patch, high forehead, hooded eyes, piercings, prominent cheekbones, full lips, braided, upturned-nosed, youthful, approachable, arched eyebrows,thin-lipped, thin-eyebrowed, birthmark, bobbed, composed, curly hair, deep-set eyes, thick-eyebrowed |
| | 40 | clean-shaven, beard, mustache, wide-eyed, thin lips, bald, glasses-wearing, freckled, almond-shaped eyes, scarred, wrinkled, soul patch, high forehead, hooded eyes, piercings, prominent cheekbones, full lips, braided, upturned-nosed, youthful, approachable, arched eyebrows,thin-lipped, thin-eyebrowed, birthmark, bobbed, composed, curly hair, deep-set eyes, thick-eyebrowed, earrings, eyebrow thickness, facial hair, goatee, heart-shaped face, long eyelashes, low forehead, monolid eyes, nasolabial folds, diamond-shaped face |
| USER | 20 | makeup, bangs, wearing eyeglasses, wearing earrings, black hair, arched eyebrows, blonde hair, red lip, gray hair, beard, wavy hair, child, bald head, smiling, double chin, wearing hat, young, man, woman, wearing necklace |

## A.2 DETAILS OF CELEBA ACCURACY EXPERIMENT

Table S8 displays binary classification results for all attributes in CelebA using both CS and HCS, comparing them to the ground truth attribute labels. By setting the threshold based on the number of positive labels for each CelebA attribute, we found that the accuracy and F1 score of HCS are superior to CS, regardless of whether we use micro or macro averaging. Additionally, we conducted experiments by setting the origin of HCS as the overall mean of both image and text means, validating that using separate text and image means is essential.

Table S7: **Attributes used for CelebA accuracy experiment**

| Attribute type | Attribute |
|---|---|
| Refined attributes | Arched_Eyebrows, Bags_Under_Eyes, Bald, Bangs, Big_Nose, Black_Hair, Blond_Hair,Brown_Hair, Chubby, Double_Chin, Eyeglasses, Goatee, Gray_Hair, Heavy_Makeup, Male, Mouth_Slightly_Open, Mustache, No_Beard, Sideburns, Smiling, Straight_Hair, Wavy_Hair, Wearing_Earrings, Wearing_Hat,Wearing_Lipstick, Wearing_Necklace, Wearing_Necktie, Young |
| All attributes | 5_o_Clock_Shadow, Arched_Eyebrows, Attractive, Bags_Under_Eyes, Bald, Bangs, Big_Lips,Big_Nose, Black_Hair, Blond_Hair, Blurry, Brown_Hair, Chubby, Double_Chin, Eyeglasses, Goatee, Gray_Hair, Heavy_Makeup, High_Cheekbones, Male, Mouth_Slightly_Open, Mustache, Narrow_Eyes, No_Beard, Oval_Face, Pale_Skin, Pointy_Nose, Receding_Hairline, Rosy_Cheeks, Sideburns, Smiling, Straight_Hair, Wavy_Hair, Wearing_Earrings, Wearing_Hat,Wearing_Lipstick, Wearing_Necklace, Wearing_Necktie, Young |

Table S8: **Accuracy from CelebA ground truth labels.** Heterogeneous CLIPScore with origin at the entire center of images and texts is seriously inferior to the one with origin at the separate center of images ($C_\mathcal{X}$) and texts ($C_\mathcal{A}$). It validates the definition of $V_a$.

| | | accuracy | f1 score(macro) | f1 score(micro) |
|---|---|---|---|---|
| All attributes | HCS (seperate) | **0.794** | **0.442** | **0.545** |
| | HCS (entire) | 0.737 | 0.312 | 0.416 |
| | CS | 0.781 | 0.392 | 0.515 |
| Refined attributes | HCS (seperate) | **0.817** | **0.519** | **0.616** |
| | HCS (entire) | 0.751 | 0.366 | 0.475 |
| | CS | 0.798 | 0.450 | 0.575 |

Table S9: **Top 40 appeared attribute in COCO validation captions.** The first and third rows represent the attributes in COCO validation captions, while the second and fourth rows represent the corresponding number of appearances of these attributes in the captions.

| man | woman | he | people | person | table | group | street | water | plate | cat | field | couple | dog | side | food | beach | bed | bathroom | road |
|---|---|---|---|---|---|---|---|---|---|---|---|---|---|---|---|---|---|---|---|
| 20262 | 9352 | 8212 | 8164 | 7196 | 6584 | 6401 | 4382 | 3741 | 3717 | 3476 | 3385 | 3301 | 3071 | 2981 | 2973 | 2731 | 2687 | 2477 | 2377 |
| grass | kitchen | skateboard | picture | road | train | building | snow | surfboard | toilet | giraffe | room | men | bunch | ball | air | bench | clock | boy | sign |
| 2346 | 2286 | 2259 | 2209 | 2165 | 2140 | 2108 | 2097 | 1968 | 1879 | 1874 | 1827 | 1819 | 1809 | 1807 | 1710 | 1630 | 1607 | 1573 | 1569 |

# B ADDITIONAL ABLATION STUDY

## B.1 NECESSITY OF SEPARATING IMAGE MEAN AND TEXT MEAN

In the main paper, we defined Heterogeneous CLIPScore as computing angles between vectors $V_x$ and $V_a$. $V_x$ is a vector from the center of images to an image in CLIP space. $V_a$ is a vector from the center of captions to an attribute in CLIP space. Table S8 quantitatively validates the effectiveness of setting the origin of $V_a$ as the center of captions ($C_\mathcal{A}$) compared to the center of images ($C_\mathcal{X}$).

## B.2 REPLACING HETEROGENEOUS CLIPSCORE WITH CLIPSCORE

We also include additional comparisons of SaD and PaD across different image injection settings with CLIPScore rather than Heterogeneous CLIPScore in Figure 5.1. Compared to the validation result with Heterogeneous CLIPScore, both results reflect a corresponding tendency: the more correlated image injected, the worse performance in the proposed metric. However, considering the quantitative effectiveness we demonstrated for Heterogeneous CLIPScore in Table S8, we highly recommend using Heterogeneous CLIPScore with proposed metrics: SaD and PaD.

## B.3 CAN SAD AND PAD ALSO CAPTURE SKIPS OF ATTRIBUTE?

We validate that SaD and PaD accurately capture the skipness of certain attributes in Table S11. Using CelebA annotation labels, we construct sets A and B with 50k images, each naturally containing 3,325 and 3,260 images with eyeglasses, respectively. As we intentionally replace images with eyeglasses in set B with images without eyeglasses, SaD and PaD deteriorated linearly with an increasing number

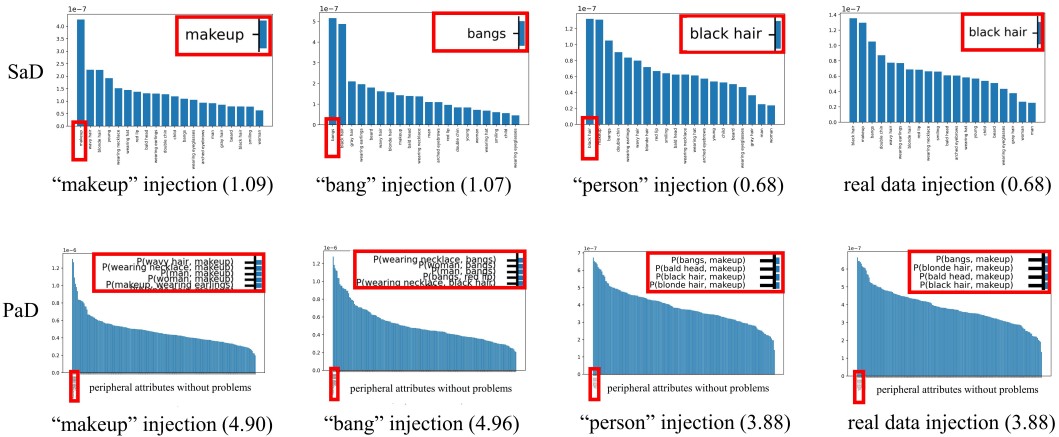

Figure S7: **Correlated images injection experiment.**

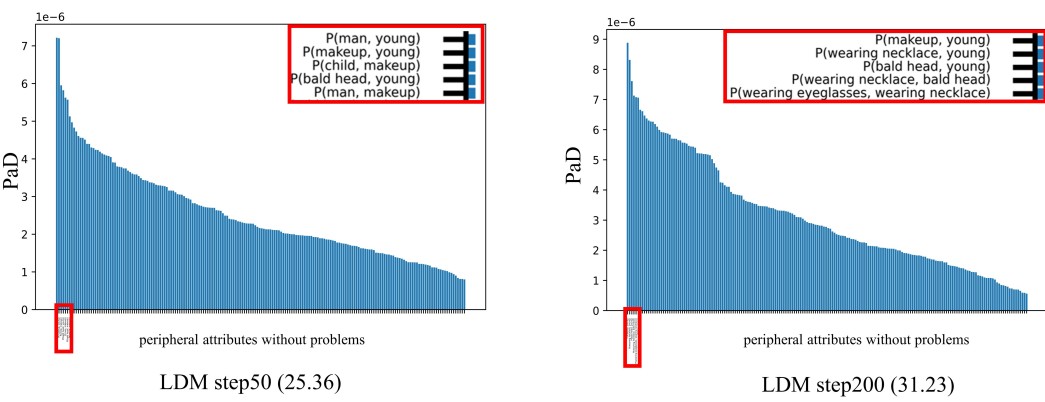

Figure S8: **PaD for LDM with different sampling timesteps.**

Table S10: **SaD and PaD scores for text-to-image models.** Stable diffusion v1.5 outperforms Stable Diffusion v2.1 in SaD and PaD regardless of the number of attributes despite being known as inferior in FID.

| | SaD | | | PaD | | |
|---|---|---|---|---|---|---|
| | $N_{\mathcal{A}} = 20$ | $N_{\mathcal{A}} = 30$ | $N_{\mathcal{A}} = 40$ | $N_{\mathcal{A}} = 20$ | $N_{\mathcal{A}} = 30$ | $N_{\mathcal{A}} = 40$ |
| SDv1.5 | **37.91** | **24.37** | **25.44** | **87.53** | **60.71** | **62.47** |
| SDv2.1 | 69.49 | 48.23 | 47.53 | 146.12 | 106.86 | 105.03 |

of replaced images, with the `eyeglasses` attribute making a more significant contribution to SaD and Pad. It demonstrates proposed metric effectively catches the skipness of some attributes, and accurately captures the distribution change of the attribute HCS probability density function.

### B.4 MORE DETAILS: TEXT-TO-IMAGE MODEL EVALUATION

We compare Stable Diffusion v1.5 and Stable Diffusion v2.1 on the COCO dataset using top-N appeared attributes in COCO validation captions (Table S9). Regardless of number of attributes, SDv1.5 outperforms SDv2.1 in SaD and PaD (Figure S14, Table S10).

### B.5 HUMAN EVALUATION

We show SaD and PaD are consistent with human judgment on the CelebA dataset. 40 participants participated in these surveys.

Table S11: **Validation result of skips experiment**

| | | | SaD | PaD | most influencing attribute for SaD |
|---|---|---|---|---|---|
| $\frac{\text{eyeglasses } 3325}{\text{total } 50000}$ | v.s. | $\frac{\text{eyeglasses } 3260}{\text{total } 50000}$ | 0.63 | 3.42 | beard |
| $\frac{\text{eyeglasses } 3325}{\text{total } 50000}$ | v.s. | $\frac{\text{eyeglasses } 2000}{\text{total } 50000}$ | 0.89 | 4.05 | **eyeglasses** |
| $\frac{\text{eyeglasses } 3325}{\text{total } 50000}$ | v.s. | $\frac{\text{eyeglasses } 1000}{\text{total } 50000}$ | 1.54 | 5.66 | **eyeglasses** |
| $\frac{\text{eyeglasses } 3325}{\text{total } 50000}$ | v.s. | $\frac{\text{eyeglasses } 3325}{\text{total } 50000}$ | 3.25 | 11.59 | **eyeglasses** |

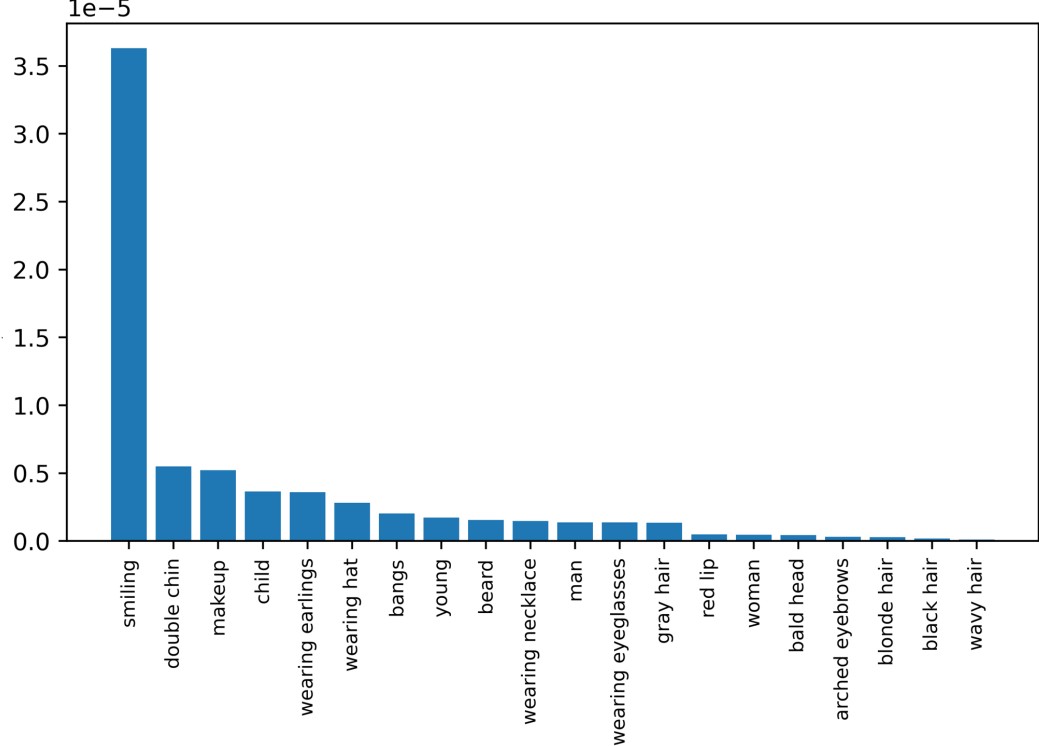

Figure S9: **SaD for Section 5.2: Discernment of PaD.**

Table S12: **Correlation between human judgements and SaD.**

| set A | set B | SaD | Human 1st (%) | Human 2nd (%) | Human 3rd (%) |
|---|---|---|---|---|---|
| strong smile | strong smile | **0.89** | **99.48** | 0 | 0.51 |
| | medium smile | 19.39 | 0 | **99.48** | 0.51 |
| | no smile | 92.46 | 0.51 | 0.51 | **98.97** |

Table S13: **Correlation between human judgements and PaD.**

| set A | set B | PaD | Human 1st (%) | Human 2nd (%) | Human 3rd (%) |
|---|---|---|---|---|---|
| r=1 | r=1 | **4.57** | **94.36** | 3.59 | 2.05 |
| | r=0 | 38.43 | 2.56 | **93.85** | 3.59 |
| | r=-1 | 117.58 | 3.08 | 2.56 | **94.36** |

**SaD**   Figure S11 shows a correlation between SaD and human judgments. We asked the participants to mark if two sets have different distribution of smile. One set is fixed as a training set with 50% `smile`. Another set varies from 0% `smile` to 100% `smile`. We used smiling and non-smiling images from CelebA ground truth labels. Meanwhile, we measure SaD between the two sets and for

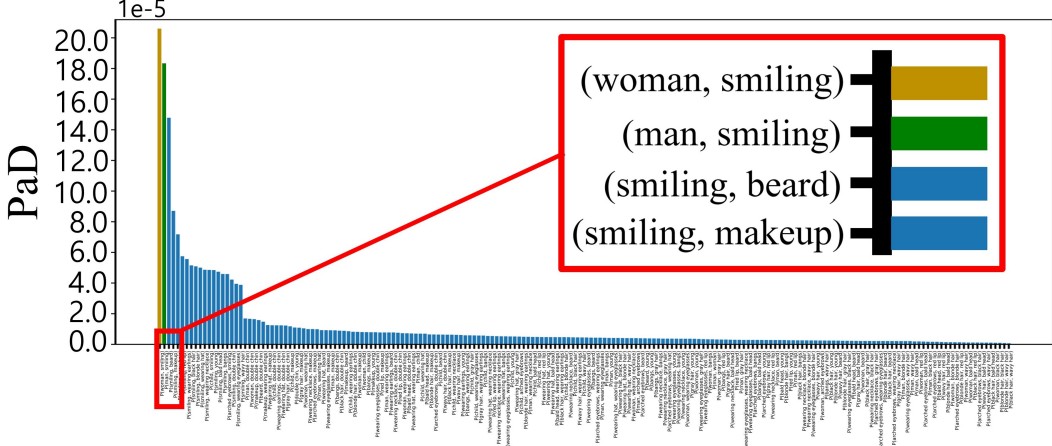

Figure S10: **PaD for Section 5.2: Discernment of PaD.**

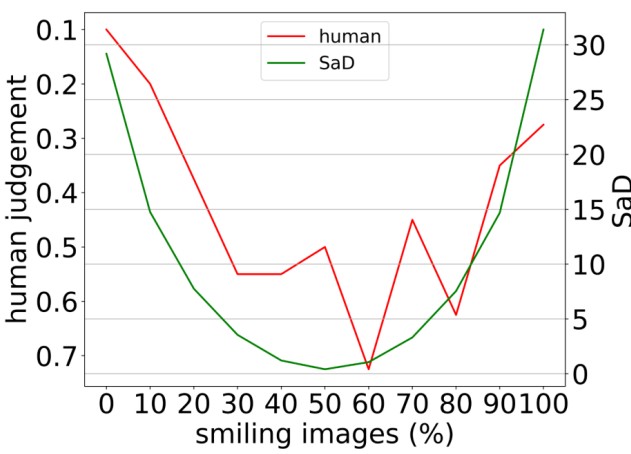

Figure S11: **Correlation between human judgements and SaD.**

comparison. Notably, both SaD and human judgement rapidly increase with increasing and decreasing smile in >80% and <30% range, respectively. Likewise, both SaD and human judgement have gentle change with same sign of slope in 30% < smile < 80% range.

**PaD**  Table B.4 shows a correlation between PaD and human judgments. Based on the given ground-truth set A, participants ranked three sets; 1) a set with strong positive correlation (r=1) 2) a set with zero-correlation (r=0) and 3) a set with strong negative correlation (r=-1).

We opt to use the correlations between `man` and `smile` and we gave five triplets to the participants to rank within the triplets. Most (about 94%) of the participants identified the rank of correlation between `man` and `smile` correctly and it aligns with PaD.

## C  MORE DETAILED RESULTS AND ANALYSIS

In this section, we provide analysis of various generative models using our metric's explicit interpretability.

**SaD**  Figure S13 shows the SaD results for StyleGAN 1, 2, 3, iDDPM, and LDM with two different step versions, StyleSwin, and ProjectedGAN. For LDM, DDIM sampling steps of 50 and 200 were used, and all numbers of the images are 50k.

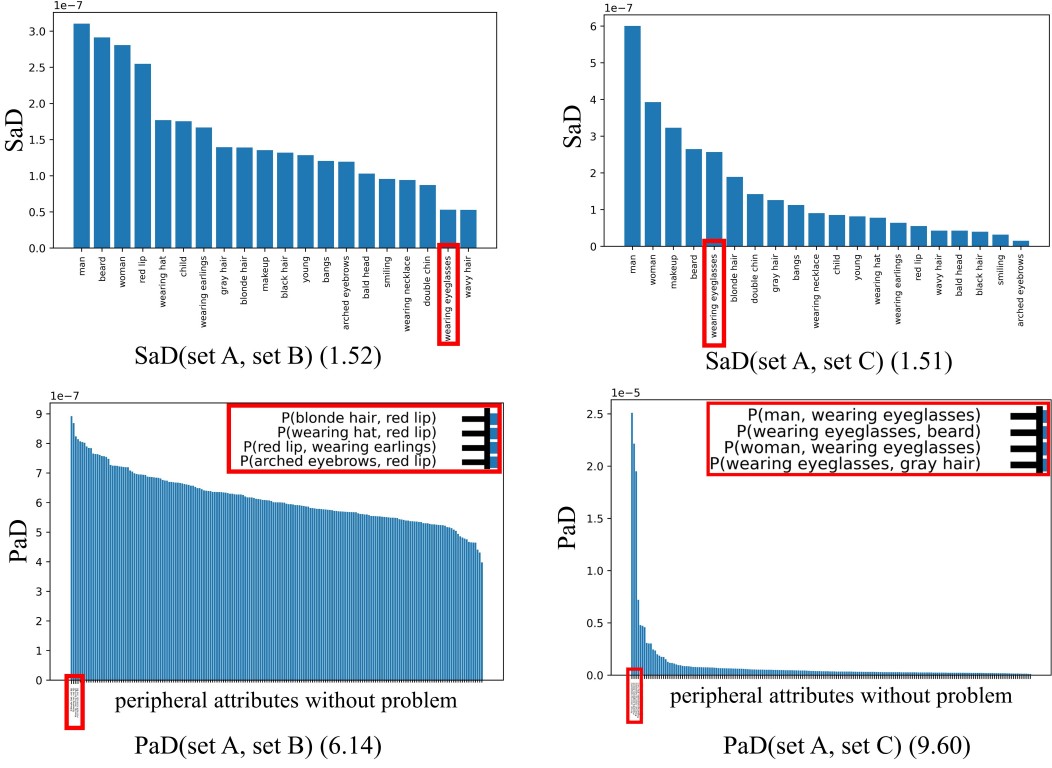

Figure S12: **Additional experiment for Section 5.2: Necessity of PaD over SaD.** In set A and set B only `men` wear `eyeglasses`, while only `women` wear `eyeglasses` in set C. PaD successfully captures pairwise relation errors between set A and set C, whereas SaD cannot.

Table S14: **Top 3 PaD pair with USER attributes on FFHQ**

|     | StyleGAN1 | StyleGAN2 | StyleGAN3 | iDDPM | LDM (50) | LDM (200) | StyleSwin | ProjectedGAN |
|-----|-----------|-----------|-----------|-------|----------|-----------|-----------|--------------|
| 1st | man &woman | arched eyebrows &makeup | red lip &makeup | arched eyebrow &makeup | man &young | makeup &young | makeup &young | man &woman |
| 2nd | child &makeup | child &makeup | arched eyebrow &makeup | woman &arched eyebrow | makeup &young | wearing necklace &young | woman &young | red lip &makeup |
| 3rd | makeup &young | man &woman | child &makeup | child &makeup | child &makeup | bald head &young | wavy hair &young | child &makeup |

SaD directly measures the differences in attribute distributions, indicating the challenge for models to match the density of the highest-scoring attributes to that of the training dataset. Examining the top-scoring attributes, all three StyleGAN models have similar high scores in terms of scale. However, there are slight differences, particularly in StyleGAN3, where the distribution of larger accessories such as `eyeglasses` or `earrings` differs. Exploring the training approach of alias-free modeling and its relationship with such accessories would be an interesting research direction.

In contrast, iDDPM demonstrates notable scores, with attributes `makeup` and `woman` showing scores over two times higher than GANs. Particularly, apart from these two attributes, the remaining attributes are similar to GANs, highlighting significant differences in the density of `woman` and `makeup`. Investigating how the generation process of diffusion models, which involves computing gradients for each pixel, affects attributes such as `makeup` and `woman` would be an intriguing avenue for future research.

For LDM, while FID improves with more timesteps, SaD gets worse. Specifically, the scores for `earrings`, `necklace`, and `young` significantly increase with 200-step results. Analyzing the influence of attributes as the number of steps increases, leading to more frequent gradient updates, would be a highly interesting research direction. Moreover, diffusion models are known to generate

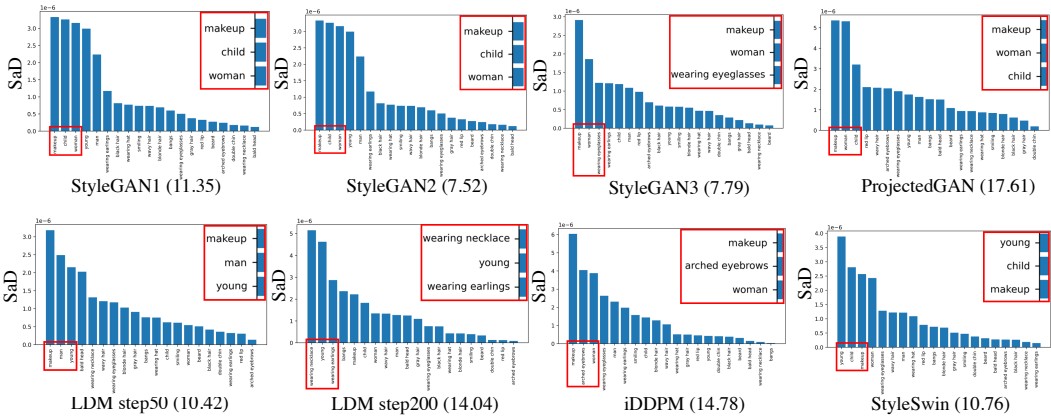

Figure S13: **SaD with USER attributes on FFHQ.**

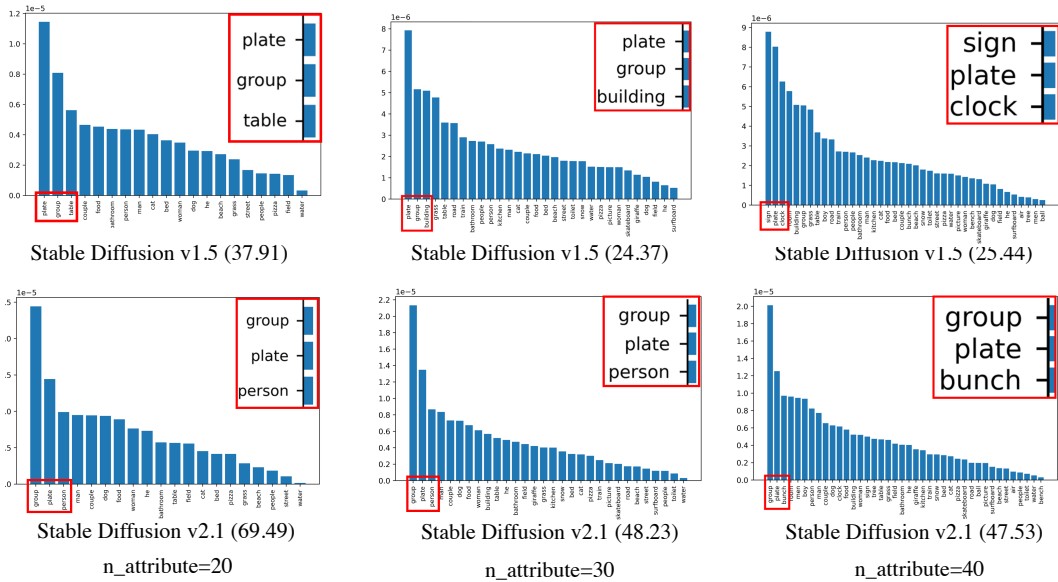

Figure S14: **SaD for text-to-image models.** Stable Diffusion v1.5 outperforms stable Diffusion v2.1 in SaD. regardless of the number of attributes despite being known as inferior in FID.

different components at each timestep. Understanding how these model characteristics affect attributes remains an open question and presents an intriguing area for exploration.

**PaD** PaD provides a quantitative measure of the appropriateness of relationships between attributes. Thus, if a model generates an excessive or insufficient number of specific attributes, it affects not only SaD but also PaD. Therefore, it is natural to expect that attribute pairs with high PaD scores will often include worst-ranking attributes in SaD. Table S14 presents the worst three attributes with the highest PaD scores, and their overall values can be found in Table 2.

PaD reveals interesting findings. Firstly, it is noteworthy that attributes related to `makeup` consistently receive high scores across all StyleGAN 1, 2, and 3 models. (Table S14) This indicates that GANs generally fail to learn the relationship between `makeup` and other attributes, making it an intriguing research topic to explore the extent of this mislearning and its underlying reasons.

In the case of iDDPM, the values for `arched eyebrows` and `makeup` are overwhelmingly higher compared to other attributes. The reasons behind this will be discussed in the following subsection.

Table S15: **Worst 3 PaD pair with shape/color attributes on LSUN Cat**

|  |  | StyleGAN1 | StyleGAN2 | iDDPM |
|---|---|---|---|---|
| color attributes | 1st | fawn fur &navy fur | fawn fur &calcico fur | tabby fur &striped fur |
|  | 2nd | fawn fur &calcico fur | fawn fur &lilac fur | dotted fur &striped fur |
|  | 3rd | lilac fur &fawn fur | lilac fur &navy fur | black fur &striped fur |
| shape attributes | 1st | tufted ears &slanted eyes | tufted ears &slanted ears | hazel eyes &long tail |
|  | 2nd | pointed ears &slanted eyes | tufted ears &white chin | Almond-shaped eyes &long tail |
|  | 3rd | slanted eyes small ears | pointed ears &white chin | long tail &wide-set eyes |

Figure S15: **SaD for LSUN Cat with color attributes.**

## C.1 COMPARING GENERATIVE MODELS WITH SPECIFIC ATTRIBUTE TYPES

In the main paper, we suppose that the distribution of color-related attributes has a harmful effect on the DMs' performances compared to shape-related attributes on the proposed metric. In this section, we analyze which specific attribute DMs are hard to generate compared to StyleGAN models.

**Color-related attributes** Figure S15 illustrates the color-related result of SaD that iDDPM fails to preserve attributes with patterns such as `striped fur` and `dotted fur`. Considering that the color in the diffusion model is largely determined by the initial noise, we suppose that creating texture patterns such as stripes or dot patterns would be challenging. This characteristic is also observed in PaD. Unlike GANs, we can observe that relationships between solid colors without patterns or textures are not among the worst 3 attributes. (Table S15)

**Shape-related attributes** SaD and PaD of Shape-related attributes were relatively lower than color-related attributes. However, the attributes that have a negative impact on the scores are different in StyleGANs and iDDPM as shown in Figure S16.

Interestingly, among the attributes that DMs struggle with, the worst two attributes, `long tail` and `tufted ears`, share the commonality of being thin and long. We speculate that this is similar to the difficulty in creating `stripes`, indicating a similar characteristic.

These conjectures also explain why `arched eyebrows` in FFHQ have a high PaD score. Arched eyebrows have a thin and elongated shape that differs from the typical eyebrow appearance. Considering the characteristics of diffusion models that struggle to create stripes effectively, we can gain insights into the reasons behind this observation.

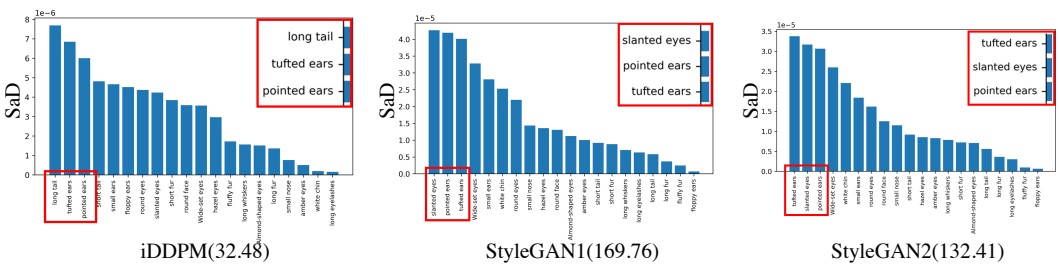

Figure S16: **SaD for LSUN Cat with shape attributes.**

