# OpenReview forum: "Attribute Based Interpretable Evaluation Metrics for Generative Models"
_ICLR.cc/2024/Conference — Submitted to ICLR 2024_

### Official Review · Reviewer_rWJQ · 2023-10-21

**Soundness:** 1 poor
**Presentation:** 2 fair
**Contribution:** 2 fair
**Rating:** 3
**Confidence:** 3

**Summary:**

The authors propose to evaluate the quality of image generation approaches by comparing the distribution of attributes (and attribute pairs) scores across the training dataset and the generated one. This is intended to capture whether the generative model has captured well the training distribution.
The requires either an auxiliary model trained for attribute detection, either ad hoc or a generic one like CLIP.

**Strengths:**

The method is well motivated.

**Weaknesses:**

1- The experiments aimed at investigating the behaviour of the approach seem to be, in their current form, inconclusive.
1.1 – In 5.1, it is not explained how the “normal images” are obtained. This prevents us from discerning whether it really is the out-of-distribution attributes that increase the scores, or simply the difference between the generated images and the normal ones.
1.2 – In 5.2, the authors intend to highlight the need of PaD over SaD. However, they do not actually compare them, with no results for SaD to be found in this section.
1.3 – The approach would be, by nature, sensitive to the quality of the attribute detector, and only attributes that are consistently visually detectable should be used, since the metrics would be a mixture of the quality of the generator and that of the detector.
2- The paper needs to be improved in terms of writing and structure.
2.1 – With respect to the writing, there are many sentences that need some improvement. Just some examples:
- “ They provide which attributes the models struggle.”
- “all SaD top-rank attributes have negative mean differences that mean SDs tend to omit some objects”
- “We infer that the color-related attributes are the inferiority of DMs”
2.2 – The mean difference (Eq 7), seems to be an element of the methodology but appears in the experiments section.
2.3 – Many of the figures (like Fig 4a) display text that is impossible to read to to its size (and when zooming in it actually becomes pixelated).
3 – Some of the comparisons could be more comprehensive. For instance, Table 4 shows no other metrics than the proposed ones.

**Questions:**

Please see the weaknesses.

---

> ### Author Response · Authors · 2023-11-14
>
> We appreciate Reviewer rWJQ for the constructive feedback, especially highlighting the sound motivation of our method.
>
> Below, we carefully address the concerns.
>
> ### 1.1. Explanation for "normal images"
> > In 5.1, it is not explained how the “normal images” are obtained. This prevents us from discerning whether it really is the out-of-distribution attributes that increase the scores, or simply the difference between the generated images and the normal ones. Please review the final version.
>
>
> Normal images and the initial 30K images are IID sampled from *the same distribution*, i.e., the training set, without repetition. It works as a controlled counterpart of adding images with OOD pairs of attributes. We added this in the revised version. Thank you for providing preciseness for our paper.
>
> ### 1.2. Necessity of PaD over SaD
> > In 5.2, the authors intend to highlight the need of PaD over SaD. However, they do not actually compare them, with no results for SaD to be found in this section.
>
>
> We respectfully remind that PaD detects correlation errors such as "babies with beard", while SaD lacks this capability *by design*.
>
> Besides, we added SaD in Appendix Figure S9. It only reveals divergence in single attribute, e.g., *smile*, while PaD reveals correlation errors.
>
> ### 1.3. Dependency on the attribute detector
> > The approach would be, by nature, sensitive to the quality of the attribute detector, and only attributes that are consistently visually detectable should be used, since the metrics would be a mixture of the quality of the generator and that of the detector
> >
>
> This is the characteristics shared by most existing metrics that use feature extractor including FID, precision, recall, LPIPS, etc. We respectfully suggest that designing metrics without dependency is an orthogonal research direction.
>
> ### 2.1. Writing improvements
>
> Thank you for helping us improve readability. Changes are summarized below.
>
> > Single-attribute Divergence (SaD) measures the divergence regarding PDFs of a single attribute. Paired-attribute Divergence (PaD) measures the divergence regard- ing joint PDFs of a pair of attributes. *They provide which attributes the models struggle*.
>
> $\rightarrow$ Single-attribute Divergence (SaD) *reveals the attributes that are generated excessively or insufficiently by measuring* the divergence of PDFs of individual attributes. Paired-attribute Divergence (PaD) *reveals such pairs of attributes by measuring* the divergence of joint PDFs of pairs of attributes.
>
>
> > *all SaD worst-rank attributes have negative mean differences that mean SDs tend to omit some objects*
>
> $\rightarrow$ *mean difference of attribute strengths are below zero. It implies that SDs tend to omit some objects rather than over-generating them.*
>
>
> > *We infer that the color-related attributes are the inferiority of DMs.*
>
> $\rightarrow$ *It implies that DMs fall short on modeling color-related attributes.*
>
>
> Besides, we have revised writings that were difficult to comprehend. Please refer to the revised version.
>
> ### 2.2.-2.3. Structure improvements
>
>
> > The mean difference (Eq 7), seems to be an element of the methodology but appears in the experiments section.
>
> Thank you for the great suggestion. We moved equation for mean difference to methology section 4.1.
>
> > Many of the figures (like Fig 4a) display text that is impossible to read to to its size (and when zooming in it actually becomes pixelated).
>
> Such figures illustrate the key observation in the enlarged insets and the content of tiny labels does not need to be read because they are peripheral attributes without problems.
>
>
> ### 3. Additional experiments
> > Some of the comparisons could be more comprehensive. For instance, Table 4 shows no other metrics than the proposed ones.
>
> In Table 4, other metrics are irrelevant because they have nothing to do with attributes. The message of Table 4 is that Stable Diffusions omit or over-generate specific attributes, and these differences vary across the versions. Nevertheless, we report them below.
>
>
> Table 4: SaD and PaD of different versions of Stable Diffusion
>
>
> |        | FID    | FID_CLIP | Precision | Recall    | Density   | Coverage  |
> |--------|--------|----------|-----------|-----------|-----------|-----------|
> | SDv1.5 | **14.748** | **8.902**    | **0.707** | 0.962     | **0.567** | **0.869** |
> | SDv2.1 | 15.203 | 10.659   | 0.663     | **0.969** | 0.414     | 0.672     |

---

> > ### Comment · Reviewer_rWJQ · 2023-11-20
> >
> > I greatly appreciate the effort of the authors in updating the paper. However, I still find the main elements of my review to be unaddressed.
> >
> > ## 1.1. Explanation for "normal images"
> > I thank the authors for clarifying that the "normal" images come from the real dataset. This reinforces my concern that this experiment doesn't allow to draw any conclusions, since the injections are of different nature: 1) biased generated images and 2) real images. In order to understand whether this difference is important, this would need to be compared against 3) unbiased generated images (since biased real images are not really an option).
> >
> > ## 1.2. Necessity of PaD over SaD
> > I don't think I follow the authors' argument here. Section 5.2 seems to be written as a comparison between SaD and PaD, in order to highlight the cases in which PaD is able to detect deviations from the original distribution that are due to pairwise interactions between attributes. It would, therefore, be sensible, and even necessary for the argument, to compare both. In case the contamination only affects the pairwise relations, and not the individual abundance of each attribute, as in described in Section 5.2, we would expect PaD to pick up the difference, while SaD may not. Currently, the result shown in Fig. 4 (and those in Fig. S9), don't allow to understand whether PaD would indeed allow to pick up this difference without assuming that we know the nature of the contamination (the fact that it relates to gender and smiling). This results are, therefore, inconclusive as well.
> >
> > ## 1.3. Dependency on the attribute detector
> > I agree with the authors that this limitation is intrinsic to the approach. I only meant that this should be acknowledged and, ideally, investigated: how does the quality of the attribute detector affect the methods?
> >
> > ## 2. Writing and structure
> > I appreciate the improvements based on the suggestions. I find that there are still some writing details that need to be polished, though, and I strongly recommend the authors to edit it carefully. For instance: in Section 5.2, "non-smiling men" and "smiling men" are written in very different styles (not clear if for a reason); in Section 5.3, the plural of phenomenon is phenomena (but also, why is iDDPM called a phenomenon?).
> >
> > With respect to the figures, I still find that they need to be substantially improved. I strongly disagree with the argument that the tiny text in the figures is fine because it "does not need to be read". If that is the case, please don't add it to the figure. Keeping with the example of Fig. 4a, not only is the label text impossible to read, but even the actual result (the bars), are tiny and hard to see, while most of the space is used for "peripheral attributes without problems". I find this quite unacceptable for a final version of a paper.

---

> > > ### Author Response · Authors · 2023-11-22
> > >
> > > We thank the reviewer for the constructive comments.
> > >
> > > ### Explanation for "normal images"
> > > >I thank the authors for clarifying that the "normal" images come from the real dataset. This reinforces my concern that this experiment doesn't allow to draw any conclusions, since the injections are of different nature: 1) biased generated images and 2) real images. In order to understand whether this difference is important, this would need to be compared against 3) unbiased generated images (since biased real images are not really an option)
> > >
> > > Thank you for the important suggestion. We agree injections from real dataset have a different nature from [DiffuseIT]. Following the suggestion, we defined a set of unbiased generated images with a prompt *person* using the same editing method [DiffuseIT].
> > >
> > > The results with *person* lead us to the **same conclusion**: **SaD and PaD do not change** when we add unbiased generated images with the same distribution as the training dataset. We updated Figure 3 and Figure S7 with *person* added, and here we report part of them. Please check them in the revised paper.
> > >
> > > Injection experiment - SaD
> > >
> > > |                  |   0   |    100    |    200    |    300    |    400    |    500    |    600    |
> > > |:----------------:|:-----:|:---------:|:---------:|:---------:|:---------:|:---------:|:---------:|
> > > | *person*  | 0.680 |   **0.680**   |  **0.680**  |   **0.680**   |   **0.677**   |   **0.677**   |   **0.680**   |
> > > | "normal images" | 0.680 | 0.686 | 0.684 | 0.680 | 0.686 |0.682 | 0.680  |
> > > | *bangs* | 0.680 |0.686 | 0.709 | 0.789 | 0.852 | 0.936 | 1.071 |
> > >
> > >
> > > [DiffuseIT] Kwon et al., Diffusion-based Image Translation using Disentangled Style and Content Representation, ICLR, 2023
> > >
> > >
> > > ### Necessity of PaD over SaD
> > > >I don't think I follow the authors' argument here. Section 5.2 seems to be written as a comparison between SaD and PaD, in order to highlight the cases in which PaD is able to detect deviations from the original distribution that are due to pairwise interactions between attributes. It would, therefore, be sensible, and even necessary for the argument, to compare both. In case the contamination only affects the pairwise relations, and not the individual abundance of each attribute, as in described in Section 5.2, we would expect PaD to pick up the difference, while SaD may not. Currently, the result shown in Fig. 4 (and those in Fig. S9), don't allow to understand whether PaD would indeed allow to pick up this difference without assuming that we know the nature of the contamination (the fact that it relates to gender and smiling). This results are, therefore, inconclusive as well.
> > >
> > >
> > > We are conducting an experiment: Settings with PaD of a pair of attributsA and B is notably high, while SaD of attribute A and attribute B is relatively low. We will report the results in a next comment as soon as possible.
> > >
> > > ### Dependency on the attribute detector
> > >
> > > >I agree with the authors that this limitation is intrinsic to the approach. I only meant that this should be acknowledged and, ideally, investigated: how does the quality of the attribute detector affect the methods?
> > >
> > > We added this discussion into Conclusion and Discussion section:
> > > >>Our metrics can be influenced by quality of attribute detector.
> > >
> > > We will conduct an investigation on the malfunctions caused by bias in attribute detector, with the results to be presented in the camera-ready version. Alternatively, exploring the impact of bias in a large model on various metrics could be a promising topic for a separate paper.

---

> > > > ### Author Response · Authors · 2023-11-22
> > > >
> > > > ### Writing and structure
> > > >
> > > > >I appreciate the improvements based on the suggestions. I find that there are still some writing details that need to be polished, though, and I strongly recommend the authors to edit it carefully. For instance: in Section 5.2, "non-smiling men" and "smiling men" are written in very different styles (not clear if for a reason); in Section 5.3, the plural of phenomenon is phenomena (but also, why is iDDPM called a phenomenon?).
> > > > Thank you for helping us improve readability. Changes are summarized below.
> > > >
> > > >
> > > > >> ForX , we gathered 20,000 *smiling men* and 20,000 *non-smiling women* using CelebA’s ground truth labels. In contrast, Y comprised 20,000 ‘non-smiling men’ and 20,000 ‘smiling women’.
> > > >
> > > > $\rightarrow$ For $\mathcal{X}$, we gathered 20,000 *smiling men* and 20,000 *non-smiling women* using CelebA’s ground truth labels. In contrast, $\mathcal{Y}$ comprised 20,000 *non-smiling men* and 20,000 *smiling women*.
> > > >
> > > >
> > > > We observe that iDDPM hardly generates accurate attribute distributions for the training set related to facial aesthetics, as a phenomenon. This word was redundant, and we revised the sentences as below.
> > > >
> > > > >> In addition, we find phenomenons such as iDDPM demonstrate notable scores, with the attribute *arched eyebrows* showing scores over two times higher than GANs in SaD, and attributes related to makeup consistently receive high scores across all StyleGAN 1, 2, and 3 models in PaD.
> > > >
> > > > $\rightarrow$ In addition, iDDPM shows notable scores, with the attribute *arched eyebrows* showing scores over two times higher than GANs in SaD, and attributes related to makeup consistently receive high scores across all StyleGAN 1, 2, and 3 models in PaD.
> > > >
> > > > Besides, we have revised writing details clearly that were difficult to comprehend before. Please refer to the revised version.
> > > >
> > > > >>  combinations like man ∩ makeup and man ∩ bangs for PaD
> > > >
> > > > $\rightarrow$ relationships that are rarely seen in training data such as (*man*, *makeup*) and (*man*, *bangs*) for PaD.
> > > >
> > > >
> > > > > With respect to the figures, I still find that they need to be substantially improved. I strongly disagree with the argument that the tiny text in the figures is fine because it "does not need to be read". If that is the case, please don't add it to the figure. Keeping with the example of Fig. 4a, not only is the label text impossible to read, but even the actual result (the bars), are tiny and hard to see, while most of the space is used for "peripheral attributes without problems". I find this quite unacceptable for a final version of a paper.
> > > >
> > > > Thanks for great suggestion. In response, we revised Fig 4a, showcasing only the most influential attribute/attribute pairs contributing to SaD and PaD. Besides, the labels of the peripheral attributes are removed from Figure 4a, S7, and S8. We have also included complete figures for SaD and PaD (Figure S9, Figure S10),  allowing for the full readability of all text labels.
> > > >
> > > >
> > > > We kindly remind the reviewer to raise the rating if our response has addressed the concerns. Thank you once again!

---

> > > > > ### Author Response · Authors · 2023-11-23
> > > > >
> > > > > ### Necessity of PaD over SaD (cont.)
> > > > >
> > > > > We thank the reviewer for waiting until our additional experiment concludes.
> > > > >
> > > > > > I don't think I follow the authors' argument here. Section 5.2 seems to be written as a comparison between SaD and PaD, in order to highlight the cases in which PaD is able to detect deviations from the original distribution that are due to pairwise interactions between attributes. It would, therefore, be sensible, and even necessary for the argument, to compare both. In case the contamination only affects the pairwise relations, and not the individual abundance of each attribute, as in described in Section 5.2, we would expect PaD to pick up the difference, while SaD may not. Currently, the result shown in Fig. 4 (and those in Fig. S9), don't allow to understand whether PaD would indeed allow to pick up this difference without assuming that we know the nature of the contamination (the fact that it relates to gender and smiling). This results are, therefore, inconclusive as well.
> > > > >
> > > > >
> > > > > We support the necessity of PaD over SaD by presenting an additional experiment as follows.
> > > > > 1. We curate three sets of images sampled from CelebA, with the identical marginal distribution of individual attributes.
> > > > >   **A**. Randomly sampled with a restriction: only *men* wear eyeglasses.
> > > > >   **B**. Randomly sampled with a restriction: only *men* wear eyeglasses (Identical to set A).
> > > > >   **C**. Randomly sampled with a restriction: only *women* wear eyeglasses (Corrupted correlation).
> > > > > 1. We measure SaD and PaD of **B** and **C** from **A**, i.e., SaD(**A**, **B**), SaD(**A**, **C**), PaD(**A**, **B**), and PaD(**A**, **C**).
> > > > > 1. Observe whether SaD captures difference due to the corrupted correlation between gender and eyeglasses.
> > > > >
> > > > > Attribute distributions of set **A** and **B**
> > > > > |       |  |  |     |
> > > > > |:-----:|:------------:|:------------:|:---:|
> > > > > |       | eyeglasses O | eyeglasses X |  marginals   |
> > > > > |  man  |    **0.1**   |      0.4     | 0.5 |
> > > > > | woman |       0      |      0.5     | 0.5 |
> > > > > |   marginals    |      0.1    |      0.9    | 1.0 |
> > > > >
> > > > >
> > > > > Attribute distribution of set **C**
> > > > > |       |  |  |     |
> > > > > |:-----:|:------------:|:------------:|:-----:|
> > > > > |       | eyeglasses O | eyeglasses X |  marginals   |
> > > > > |  man  |       0      |      0.5     | 0.5 |
> > > > > | woman |    **0.1**   |      0.4    | 0.5 |
> > > > > |   marginals    | 0.1         | 0.9         | 1.0 |
> > > > >
> > > > > Note that the marginals are the same across **A**, **B**, and **C**. Hence we expect SaD to be consistent. Furthermore, we expect PaD to be different because the correlation between gender and eyeglasses is corrupted.
> > > > >
> > > > > |                      |  |        |  |
> > > > > |:--------------------:|:----:|:--------:|:-------------------:|
> > > > > |                      |  SaD |     PaD   | PaD(man&eyeglasses) |
> > > > > | between sets **A** and **B** | 1.52 |   6.14   |        5.68      |
> > > > > | between sets **A** and **C** | 1.51 |  **9.60** |      **251.34**     |
> > > > >
> > > > >
> > > > >
> > > > > Indeed, SaD(**A**, **B**) $\simeq$ SaD(**A**, **C**) while PaD(**A**, **B**) << PaD(**A**, **C**). In addition, PaD correctly reveals the most influential pair of attributes (man & eyeglasses) by 26$\times$ higher PaD than the mean PaD.
> > > > >
> > > > > This highlights the effectiveness of PaD in capturing errors in pairwise relations, and the necessity of employing PaD for comprehensive model analysis.
> > > > >
> > > > > *PaD is a new evaluation metric, using the novel concept of paired attributes and regarding their relationships, and is also interpretable.* In our humble opinion, this is of great interest to the ICLR community.
> > > > >
> > > > > We added full SaD and PaD in Figure S12. Please refer to the revised version.
> > > > > We will incorporate this discussion into the main paper of the final camera-ready version.

---

### Official Review · Reviewer_B3SF · 2023-10-30

**Soundness:** 4 excellent
**Presentation:** 4 excellent
**Contribution:** 4 excellent
**Rating:** 8
**Confidence:** 5

**Summary:**

- This work proposes two new evaluation metrics, named Single-attribute Divergence (SaD) and Paired-attribute Divergence (PaD), that measure the divergence of a set of generated images from the training set.
- SaD measure the divergence of the marginal PDF of a single attribute, while PaD measures the divergence of the joint distribution of two attributes between a set of generated and ground truth images.
- To measure the attribute strengths of an image, authors propose Heterogenous CLIP score which is based on heterogenous starting points. This formulation avoids the narrow range of values that result from a CLIP score achieving scores that are unrestricted and flexible.
- Finally, authors perform experiments comparing some popular GAN and Diffusion models, revealing some interesting properties, which can be attributed to the interpretability of the proposed metrics.

**Strengths:**

- Evaluating generative models is a very important area of research given the exponential progress achieved recently in this space. Researchers have identified several shortcomings with existing automatic evaluation methods of generative models which requires more analysis and research. This work tackles a very important research problem and proposes simple and effective evaluation metrics for generative models.
- Authors identify an important shortcoming of CLIP similarity score, a popular score used for evaluating conditioned generative models, and propose an alternative which is more interpretable.
- The SaD and PaD metrics are theoretically well grounded and interpretable, unlike existing automatic evaluation metrics, making it a good diagnostic tool as well.
- Authors show some interesting preliminary analysis that agree with the general consensus of the public (SD 1.5 > SD 2.1).

**Weaknesses:**

- My major concern with the proposed metrics is with respect to resolution. How does the difference in absolute values translate to actually seeing a difference in the generations. For example, in [1] authors identify, with the help of human evaluation, that the resolution of FVD is 50, i.e. a human rater can tell the difference between generations of two models if their corresponding FVD scores differ by atleast 50 points.
- How does SaD or PaD correlate with human evaluation? Do humans agree that the attributes identified by SaD and PaD are indeed misrepresented in the generations of the model?


[1] Thomas Unterthiner, Sjoerd van Steenkiste, Karol Kurach, Raphaël Marinier, Marcin Michalski, Sylvain Gelly, FVD: A new Metric for Video Generation, ICLRW 2019

**Questions:**

- Do these metrics correlate well with any existing evaluation metrics?
- Do these metrics work with any other features than CLIP features? For example, for textures, do features from DINO/SAM (get reference features from some texture dataset and compare to features of generated images) work as well or is the joint embedding of CLIP necessary for the success of SaD and PaD.
- Can these metrics be used to measure quality and faithfulness of text to image generative models? If so, how would one go about that?

**Details Of Ethics Concerns:**

I do not foresee any immediate ethical concerns pertaining to this work.

---

> ### Author Response · Authors · 2023-11-14
>
> We appreciate Reviewer B3SF for the constructive feedback, highlighting our strengths in
> 1. Identifying shortcomings in existing evaluation methods
> 2. Proposing a more interpretable alternative to CLIP similarity
> 3. Theoretically grounded and interpretable SaD and PaD
> 4. Preliminary analysis aligning with public consensus
>
>
> Below, we carefully address the concerns.
>
>
> ### Resolution of the proposed metrics
> > My major concern with the proposed metrics is with respect to resolution. How does the difference in absolute values translate to actually seeing a difference in the generations. For example, in [1] authors identify, with the help of human evaluation, that the resolution of FVD is 50, i.e. a human rater can tell the difference between generations of two models if their corresponding FVD scores differ by atleast 50 points.
>
> Thank you for the nice suggestion.
>
> Resolution of FVD can be measured by a few samples because most users have common sense for high quality video. On the other hand, SaD and PaD require users to assess the distribution of attributes in the training set and the generated set. As it is prohibitive, to make the problem easier, we will ask users to rank the four sets of generated images from LDM-20/-50/-100/-200 which are used in Table 2. We expect the users to be able to tell the difference between sets with large SaD/PaD gaps and vice versa.
>
>
> |  | SaD | PaD |
> | -------- | -------- | -------- |
> | LDM_20steps     | 21.07     | 50.06     |
> | LDM_50steps     | 10.42     | 25.36     |
> | LDM_100steps     | 11.91     | 27.26     |
> | LDM_200steps     | 14.04     | 30.71     |
>
> We will share the results as soon as the user study is completed.
>
>
> ### Correlation with human judgements
> > How does SaD or PaD correlate with human evaluation? Do humans agree that the attributes identified by SaD and PaD are indeed misrepresented in the generations of the model?
>
> Thank you for the nice suggestion. We will conduct two user studies with toy experiments as follows.
>
> With the training set having a medium *smile*, we will ask users to rank three sets with strong, medium, and no smile. We expect the rank to agree to SaD.
>
> With the training set having a strong positive correlation between *man* and *smile*, we will ask users to rank three sets with strong positive, no, and strong negative correlations. We expect the rank to agree to PaD.
>
> We will share the results as soon as the user study is completed.
>
>
> ### Correlation with existing metrics
> > Do these metrics correlate well with any existing evaluation metrics?
>
>
> Frechet Inception Distance[FID][FID_clip] shows a consistent trend with our metrics across most generative models, while other metrics[Precision&Recall][Density&Coverage] showed no correlations with proposed metrics in Experiment 5.2.
>
> We found notable differences in PaD with FIDs, particularly revealing significant penalties for [ProjectedGAN]. We attribute ProjectedGAN's inferior performance in PaD to its ignorance to attribute correlations, generating 'babies with beards'.
>
> [FID] Heusel et al., Gans trained by a two time-scale update rule converge to a local nash equilibrium, NeurIPS, 2017
>
> [FID_CLIP] Kynkäänniemi et al., The Role of ImageNet Classes in Fréchet Inception Distance, ICLR, 2023
>
> [Precision&Recall] Sajjadi et al., Assessing generative models via precision and recall. NeurIPS, 2018
>
> [Density&Coverage] Naeem et al., Reliable fidelity and diversity metrics for generative models, ICML, 2020
>
>
> ### Other feature extractors
> > Do these metrics work with any other features than CLIP features? For example, for textures, do features from DINO/SAM (get reference features from some texture dataset and compare to features of generated images) work as well or is the joint embedding of CLIP necessary for the success of SaD and PaD.
>
> DINO/SAM with texture datasets would work but they would measure the strength of textures instead of attributes. It is a natural extension of our metrics when a user wants to evaluate the distribution of textures in the generated images. This generalization is a virtue of our metric. Using multi-modal feature extractors such as [ImageBind] also could be an interesting research topic. We added these discussion in the revised version.
>
> [ImageBind] ImageBind_ One Embedding Space To Bind Them All, Rohit Girdhar et al., CVPR2023, https://imagebind.metademolab.com/
>
>
>
> ### Evaluating text-to-image model
> > Can these metrics be used to measure quality and faithfulness of text to image generative models? If so, how would one go about that?
>
>
> _1) We are afraid but our metrics are not for measuring quality. 2) Heterogeneous CLIP score better measures faithfulness to the text than the original CLIP. 3) For the text-to-image task, SaD and PaD measure the divergence of the generated images from the training images regarding the attributes in the text prompts. Rather than the text prompts, SaD and PaD consider the training images as the reference.

---

> > ### Comment · Reviewer_B3SF · 2023-11-15
> > **Response to Author's comments**
> >
> > I would like to thank the authors for their detailed response.
> > 1. How was the correlation with other metrics measured?
> > 2. Regarding resolution, the human evaluation can be done within each attribute as well right? Show users two images with "smile" attribute which get scores within a range. You can choose different range of values and decide at which point, humans can stop telling the difference.

---

> > > ### Author Response · Authors · 2023-11-16
> > >
> > > Thank you for your response.
> > >
> > > > How was the correlation with other metrics measured?
> > >
> > >
> > > We employ [Spearman's rank correlation coefficient] and [Kendall rank correlation coefficient] to assess relation with other metrics. We observed moderate correlation between our metrics and FIDs (coefficients > 0.4), while we found very weak correlation with other metrics except Density.
> > >
> > >
> > >
> > > |           | Spearman correlation | Spearman correlation | Kendall correlation | Kendall correlation |
> > > |:---------:|:--------------------:|:--------------------:|:-----------------------:|:-----------------------:|
> > > |           |          SaD         |          PaD         |           SaD           |           PaD           |
> > > |    FID    |       **0.54**       |       **0.54**       |         **0.42**        |         **0.42**        |
> > > |  FID_CLIP |       **0.50**       |       **0.50**       |         **0.42**        |         **0.42**        |
> > > | Precision |         0.11         |         0.11         |          -0.08          |          -0.08          |
> > > |   Recall  |         0.07         |         0.07         |           0.07          |           0.07          |
> > > |  Density  |         0.46         |         0.46         |           0.34          |           0.34          |
> > > |  Coverage |         0.06         |         0.06         |          -0.05          |          -0.05          |
> > >
> > > [Spearman's rank correlation coefficient] Spearman et al. The proof and measurement of association between two things, 1904
> > >
> > > [Kendall rank correlation coefficient] Kendall et al. A New Measure of Rank Correlation, 1938
> > >
> > >
> > >
> > >
> > >
> > >
> > > >Regarding resolution, the human evaluation can be done within each attribute as well right? Show users two images with "smile" attribute which get scores within a range. You can choose different range of values and decide at which point, humans can stop telling the difference.
> > >
> > > Thanks for the great suggestion, and we will conduct user study as follows.
> > >
> > >
> > > In one set, 50% of images will have "smile", and other sets will have varying proportions of "smile" images. We will check which point human telling the difference and corresponding point's SaD and PaD.
> > >
> > >
> > > Thanks again for great suggestion.

---

> > > > ### Comment · Reviewer_B3SF · 2023-11-16
> > > > **Response to authors**
> > > >
> > > > I thank the authors for their detailed response.
> > > > - It makes sense since the baseline metrics are more generic and do not explicitly measure the quantities measured by SaD and PaD.
> > > > - Thank you for taking the suggestion. That would be interesting thing to try.
> > > > - One thing I feel which is not emphasized strongly is why do we need both PaD and SaD. Authors show an experiment that PaD can circumvent some issues of SaD. Then why not just replace SaD with PaD? I think authors should emphasize these points much stronger in the next version.

---

> > > > > ### Author Response · Authors · 2023-11-17
> > > > >
> > > > > We deeply appreciate your valuable and construct comments.
> > > > >
> > > > > > It makes sense since the baseline metrics are more generic and do not explicitly measure the quantities measured by SaD and PaD.
> > > > >
> > > > > Thanks again for recognizing our strength: SaD and PaD explicitly reveal the problematic attributes, while existings do not.
> > > > >
> > > > >
> > > > > > Thank you for taking the suggestion. That would be interesting thing to try.
> > > > >
> > > > > We really appreciate your suggestion. We will share the result by 11/20.
> > > > >
> > > > > >One thing I feel which is not emphasized strongly is why do we need both PaD and SaD. Authors show an experiment that PaD can circumvent some issues of SaD. Then why not just replace SaD with PaD? I think authors should emphasize these points much stronger in the next version.
> > > > >
> > > > > Thank you for the constructive suggestion.
> > > > >
> > > > > PaD of smile and man alone cannot indicate specific divergence among multiple factors:
> > > > > * Distribution of smile differs.
> > > > > * Distribution of man differs.
> > > > > * Joint distribution of smile and man differs.
> > > > >
> > > > > Hence, SaD should be accompanied to provide to associate or rule out single attribute divergence. If SaDs regarding smile and man are low, divergence of their joint distribution is the reason for high PaD. Otherwise, an attribute with high SaD is the pitfall of the model.
> > > > >
> > > > > We just have emphasized SaD is necessary for a comprehensive analysis of the model, alongside PaD, into the revised version.

---

> > > > > > ### Author Response · Authors · 2023-11-22
> > > > > >
> > > > > > We conducted two user studies to examine the correlation between our metrics and human judgements. 40 participants answered these surveys.
> > > > > >
> > > > > > ## About resolution
> > > > > > We asked the participants to mark if two sets have different distribution of smile. One set is fixed as a training set with 50% smile. Another set varies from 0% smile to 100% smile. We used smiling and non-smiling images from CelebA ground truth labels.
> > > > > >
> > > > > > Meanwhile, we measure SaD between the two sets and for comparison.
> > > > > >
> > > > > > |               number of smiling images           |   0%  |  10%  |  20% |  30% |  40% |  50% |  60% |  70% |  80% |  90%  |  100% |
> > > > > > |:------------------------:|:-----:|:-----:|:----:|:----:|:----:|:----:|:----:|:----:|:----:|:-----:|:-----:|
> > > > > > | **human feels different (%)** |   90.0  |   80.0  |  62.5  |  45.0  |  45.0  |  50.0  |  27.5  |  55.0  |  37.5  |   65.0  |   62.5  |
> > > > > > |          **SaD**         | 39.18 | 14.75 | 7.76 | 3.55 | 1.21 | 0.41 | 1.06 | 3.31 | 7.54 | 14.68 | 31.39 |
> > > > > >
> > > > > > Notably, both SaD and human judgement rapidly increase with increasing and decreasing smile in >80% and <30% range, respectively. Likewise, both SaD and human judgement have gentle change with same sign of slope in 30% < smile < 80% range.
> > > > > >
> > > > > >
> > > > > > ## About human judgement correlation
> > > > > > **For Single-attribute Divergence (SaD)**, based on the given ground-truth set A, participants ranked three sets;
> > > > > >
> > > > > > [B-1] a set with strong smile.
> > > > > >
> > > > > > [B-2] a set with medium smile.
> > > > > >
> > > > > > [B-3] a set with no smile.
> > > > > >
> > > > > > We opt to measure the intensity of smile in images, and we gave five triplets to the participants to rank within the triplets.
> > > > > >
> > > > > >
> > > > > > |     set A    |     set B    |  SaD  | Human 1st (%) | Human 2nd (%) | Human 3rd (%) |
> > > > > > |:------------:|:------------:|:-----:|:-------------:|:-------------:|:-------------:|
> > > > > > | strong smile | [B-1] strong smile |  0.89 |   **99.48**   |       0       |      0.51     |
> > > > > > |              | [B-2] medium smile | 19.39 |       0       |   **99.48**   |      0.51     |
> > > > > > |              |  [B-3]  no smile   | 92.46 |      0.51     |      0.51     |   **98.97**   |
> > > > > >
> > > > > > Most of (about 99%) of participants identified the rank of intensity of "smile" correctly, and it aligns with SaD.
> > > > > >
> > > > > > *We conclude that SaD and human judgement are aligned.*
> > > > > >
> > > > > > **For Paired-attribute Divergence (PaD)**, based on the given ground-truth set A, participants ranked three sets;
> > > > > >
> > > > > > [B-1] a set with strong positive correlation (**r=1**)
> > > > > >
> > > > > > [B-2] a set with zero-correlation (**r=0**)
> > > > > >
> > > > > > [B-3] a set with strong negative correlation (**r=-1**).
> > > > > >
> > > > > > We opt to use the correlations between "man" and "smile" and we gave five triplets to the participants to rank within the triplets.
> > > > > >
> > > > > > | set A                | set B | PaD     | Human 1st (%) | Human 2nd (%) | Human 3rd (%) |
> > > > > > |:---------------------:|:------:|:---------:|:-------------:|:-------------:|:-------------:|
> > > > > > | r=1                 | [B-1] r=1  | 4.57| **94.36** | 3.59 | 2.05            |
> > > > > > |                     | [B-2] r=0  | 38.43   | 2.56 | **93.85** | 3.59                  |
> > > > > > |                     | [B-3] r=-1 | 117.58  | 3.08 | 2.56 | **94.36**                  |
> > > > > >
> > > > > > Because participants ranked three sets, we report Human judgements as 1st, 2nd, and 3rd.
> > > > > >
> > > > > >
> > > > > > Most (about 94%) of the participants identified the rank of correlation between "man" and "smile" correctly and it aligns with PaD.
> > > > > >
> > > > > > *We conclude that PaD is aligned with human judgement.*
> > > > > >
> > > > > > We will update these with other attributes/pair of attributes for the camera-ready version. Please understand that we could have work with only one attribute/pair of attribute due to the short period.

---

> > > > > > > ### Comment · Reviewer_B3SF · 2023-11-22
> > > > > > > **Response to authors**
> > > > > > >
> > > > > > > I thank the reviewers for their detailed response. After carefully looking through the results I am convinced and I believe the authors have addressed my questions sufficiently. However, the authors need to carefully plan how to add this analysis to the main paper which I believe requires significant effort. I vote to keep my rating with the understanding that authors will add this and other discussions raised by other reviewers in the paper.

---

> > > > > > > > ### Author Response · Authors · 2023-11-23
> > > > > > > >
> > > > > > > > We would like to express our gratitude once again to the reviewer for the valuable remarks and unwavering support of our work.

---

### Official Review · Reviewer_7B16 · 2023-10-31

**Soundness:** 3 good
**Presentation:** 2 fair
**Contribution:** 2 fair
**Rating:** 5
**Confidence:** 4

**Summary:**

This paper proposes two evaluation metrics, single-attribute divergence and paired-attribute divergence, to measure the divergence of a set of generated images with respect to the distribution of attribute strengths. The proposed metrics are defined based on heterogeneous CLIPScore, an enhanced measure from CLIPScore. THe metrics are verified on a few generative models, including PrpjectedGAN and diffusion models, to show the effectiveness and explainability.

**Strengths:**

- This paper is clearly written and easy to follow.
- Evaluation of generative models is a crucial problem which will attract wide research interest.
- Defining evaluation metrics based on attributes to measure the divergence between image sets is a novel and reasonable idea.

**Weaknesses:**

- The overall contribution is incremental, though the research motivation is fairly clear and reasonable.
  - The heterogeneous CLIPScore is a simple extension from CLIPScore by using the centralized encodings.
  - The proposed SaD and PaD are straightforward to measure the divergences of single and paired attributes.

- As for the interpretability, I have some concerns:
  -  For me, the interpretability comes from the attributes, which are obtained from annotation or large models. So the interpretability of the evaluation metrics are limited by the set of attributes.
  - Interpretation based on attributes is only one possible solution, which may be not complete or accurate to characterize the capability of generative models.

- The attributes selection methods are somewhat simple. Little insight can be gained from this process.
  -- What if the attributes are biased due to biased annotations or large models?

**Questions:**

Please refer to "weaknesses" part for my concerns.

---

> ### Author Response · Authors · 2023-11-14
>
> We appreciate Reviewer 7B16 for the constructive feedback, highlighting our strengths in
> 1. Clear writing
> 2. Addressing the crucial issues in the evaluation
> 3. Novelty and soundness of the proposed metrics regarding attributes to measure the divergence between image sets
>
> Below, we carefully address the concerns.
>
>
> ### Compared to the motivation being clear and reasonable, Heterogeneous CLIPScore, SaD, and PaD are too simple.
> >The overall contribution is incremental, though the research motivation is fairly clear and reasonable.
> The heterogeneous CLIPScore is a simple extension from CLIPScore by using the centralized encodings.
> The proposed SaD and PaD are straightforward to measure the divergences of single and paired attributes.
>
> We sincerely appreciate recognizing our motivation.
> We respectfully suggest that *it is desirable to resolve problems using simple solutions because they are easy to adopt in future research*. Especially, *our solutions are helpful for the research community and practitioners* as follows:
>
> - Heterogeneous CLIPScore indicates the presence of attributes in a given image, while CLIPScore does not.
> - Single and Paired attribute Divergence (SaD and PaD) reveal the problematic attributes, while existing metrics do not.
>
> ### Interpretability is limited by the set of attributes.
>
> > As for the interpretability, I have some concerns:
> For me, the interpretability comes from the attributes, which are obtained from annotation or large models. So the interpretability of the evaluation metrics is limited by the set of attributes.
> Interpretation based on attributes is only one possible solution, which may be not complete or accurate to characterize the capability of generative models.
>
> Thank you for the constructive feedback. Indeed the interpretability of our metrics depends on the choice of attributes. As different users want different capabilities of the generative models, our metrics provide *customizable* measures that capture attributes designated by users. It would be beneficial to extend our work with different encoders for different modalities other than textual attributes as stated by reviewer B3SF. We added this discussion in the revised version.
>
>
>
> ### What if the attributes are biased?
> > The attributes selection methods are somewhat simple. Little insight can be gained from this process. -- What if the attributes are biased due to biased annotations or large models?
>
> Large models or annotations are merely options to ease users' burden. If a user cares about fairness, one can choose the target attributes by oneself. *Resolving the bias of annotations or large models is an orthogonal research topic.* We respectfully suggest that such bias should not be a reason for rejecting this paper. Nevertheless, we agree that we should be careful not to be biased in choosing the target attributes. This discussion is in Appendix A.1. and we added it in the main paper as well in the revised version.

---

> ### Author Response · Authors · 2023-11-20
>
> We appreciate with your thoughtful comments. Could you check our response? We will be happy to answer follow-up questions if any.

---

> > ### Comment · Reviewer_7B16 · 2023-11-23
> > **Thanks for the authors' response**
> >
> > Thanks for the effort in replying my comments and updating the submission. My concerns are partially addressed.

---

### Author Response · Authors · 2023-11-23
**Final remarks by authors**

We are deeply grateful to the reviewers for their constructive feedback. We've incorporated review feedback into our manuscript and we assure further refinement. Thank you for an excellent review process!

---

### Meta-Review · Area_Chair_WpvS · 2023-12-13

**Metareview:**

This paper proposes the evaluation of generative models via two attribute-based approaches: single-attribute divergence and paired-attribute divergence, which is incorporated into a heterogeneous CLIP score. The proposed metric reveals new properties of existing methods.

Strengths: The paper proposes relatively simple ways to extend a widely accepted metric that help the metric become more interpretable. Generative evaluation is a very important problem and needs to be studied in much more depth.

Weaknesses: It is unclear how the proposed metrics correspond to human evaluation and other evaluation (the authors do provide some results in the discussion period, however the results are not too clearly conclusive). While the authors do make a very good point about the customization of attributes for the proposed evaluation, the reviewers do bring up the importance of defining these attributes more carefully - perhaps this is something that the paper must address: what are acceptable attributes (if detected, with what confidence) for consideration under the metric.

Finally, the authors do add a lot of new results thanks to the discussion, which could be useful for making the contribution stronger.

**Justification For Why Not Higher Score:**

The absence of human evaluation makes the work somewhat hard to evaluate - what exactly is the metric capturing and how is it consistent with the knowledge we already have about the models?

**Justification For Why Not Lower Score:**

n/a

---

### Decision · Program_Chairs · 2024-01-16

Reject